# Task-dependent recurrent dynamics in visual cortex

**Satohiro Tajima[1,2]\*, Kowa Koida[3], Chihiro I Tajima[4], Hideyuki Suzuki[5], Kazuyuki Aihara[6,7], Hidehiko Komatsu[7,8]**

[1]Department of Basic Neuroscience, University of Geneva, Geneva, Switzerland; [2]JST PRESTO, Japan Science and Technology Agency, Kawaguchi, Japan; [3]EIIRIS, Toyohashi University of Technology, Toyohashi, Japan; [4]Graduate School of Information Science and Technology, University of Tokyo, Tokyo, Japan; [5]Department of Information and Physical Sciences, Graduate School of Information Science and Technology, Osaka University, Suita, Japan; [6]Institute of Industrial Science, University of Tokyo, Tokyo, Japan; [7]National Institute for Physiological Sciences, Okazaki, Japan; [8]Brain Science Institute, Tamagawa University, Machida, Japan

**Abstract** The capacity for flexible sensory-action association in animals has been related to context-dependent attractor dynamics outside the sensory cortices. Here, we report a line of evidence that flexibly modulated attractor dynamics during task switching are already present in the higher visual cortex in macaque monkeys. With a nonlinear decoding approach, we can extract the particular aspect of the neural population response that reflects the task-induced emergence of bistable attractor dynamics in a neural population, which could be obscured by standard unsupervised dimensionality reductions such as PCA. The dynamical modulation selectively increases the information relevant to task demands, indicating that such modulation is beneficial for perceptual decisions. A computational model that features nonlinear recurrent interaction among neurons with a task-dependent background input replicates the key properties observed in the experimental data. These results suggest that the context-dependent attractor dynamics involving the sensory cortex can underlie flexible perceptual abilities.

\*For correspondence: satohiro. tajima@gmail.com

**Competing interests:** The authors declare that no competing interests exist.

## Introduction

Animals are able to adapt their behavior flexibly depending on task contexts, even when the physical stimuli presented to them are identical. The physiological mechanisms underlying this flexible translation of sensory information into behaviorally relevant signals are largely unknown. Recent studies indicate that context-dependent behavior is accounted for by adaptive attractor-like dynamics in the prefrontal areas (*Mante et al., 2013*; *Stokes et al., 2013*), which associate sensory representation with behavioral responses depending on task contexts (*Freedman et al., 2001*, *2002*, *2003*; *Wallis et al., 2001*; *Wallis and Miller, 2003*; *Meyers et al., 2012*). In contrast to the prefrontal cortex, the visual areas have been suggested to show no or only modest task-related modulations of neural responses (*Sasaki and Uka, 2009*; *McKee et al., 2014*). This supports the view that sensory information is processed sequentially across the cortical hierarchy; that is, the physical properties of stimuli are encoded by the sensory cortex, and read out by the higher areas such as the prefrontal cortex.

An alternative to this sequential processing model is a view that the sensory cortex is dynamically involved in the neural mechanisms for the flexible sensory-action association. Unlike the former

model, the latter does not assume a strong differentiation between sensory and higher areas, which is described in the 'encoding-vs.-readout' framework, but allows the decision process to arise from the mutual interactions among them. In particular, assuming the involvement of sensory areas in the task-dependent behavior predicts that the neural representations in those areas are modulated by task contexts. Indeed, some studies report that neurons in the sensory areas can change their activities depending on task demands (*Koida and Komatsu, 2007*; *Mirabella et al., 2007*; *Brouwer and Heeger, 2013*). For example, it is reported that performing a color categorization task modulates the neural responses to color stimuli in the ventral visual pathway, including macaque inferior temporal (IT) cortex (*Koida and Komatsu, 2007*) and human V4 and VO1 (*Brouwer and Heeger, 2013*).

However, no clear consensus has been reached on the functional interpretations of those context-dependent sensory modulations (i.e., the effects of task contexts that alter the sensory neural responses). Some researchers suggest that the task-dependent modulation of neural activities could reflect multiple confounding factors (*Sasaki and Uka, 2009*). For example, although the task demands can modulate the neuronal response amplitudes in the IT cortex (*Koida and Komatsu, 2007*), the response amplitudes in individual neurons could be affected by the changes in arousal levels (*Greenberg et al., 2008*), visual awareness (*Lamme et al., 1998*, *2002*), task difficulty (*Chen et al., 2008*) and feature-based attention (*Treue and Martínez Trujillo, 1999*; *Kastner and Ungerleider, 2000*; *Reynolds and Heeger, 2009*).

To understand the functions and mechanisms of the task-dependent modulations in the sensory neurons, we need to elucidate the structures of collective dynamics in the neural population—in particular, the dynamical structures reflecting the perceptual functions to accomplish the tasks. To this end, in the present study we analyze the spatiotemporal structures of collective neural activity recorded from the macaque IT cortex during context-dependent behavior. To focus on functional aspects of the collective dynamics, we first characterize the evolution of neuronal states within a perceptual space that is reconstructed from the neural population activities. The analysis reveals a task-dependent dynamics of sensory representation in the IT neurons, demonstrating the emergence of discrete attractors during categorical perceptions. Moreover, those attractor dynamics are found to reflect adaptive information processing and explain behavioral variabilities. Finally, through a data analysis and a computational modeling, we suggest a potential mechanism in which the task-dependent attractor structures emerge from a bifurcation in recurrent network dynamics among the sensory and downstream areas.

## Results

We analyzed the responses of color-selective neurons recorded in the macaque IT cortex, which change their activities depending on the task demands (*Koida and Komatsu, 2007*). In the experiments, the monkeys made saccadic responses based on either of two different rules (categorization or discrimination) that associate the stimulus with different behavior. In both tasks, the monkeys were presented a sample color stimulus for 500 ms. In the categorization task, the monkeys then classified the sample color into one of two color categories, 'Red' or 'Green' (*Figure 1a*). In the discrimination task (also known as 'matching to sample'), the monkeys discriminated precise color differences by reporting which of two choice stimuli was the same color as the sample stimulus (*Figure 1b*). We then analyzed the neural responses to the sample colors in the two tasks—where the visual stimuli were physically identical between those tasks.

A previous study reported that about 64% of recorded IT cells changed their response magnitudes significantly depending on the task demands (*Figure 1c*) (*Koida and Komatsu, 2007*). Although the earlier reports have demonstrated that the modulations in individual sensory neurons could be correlated to the hypothetical models that encode categorical information (*Koida and Komatsu, 2007*; *Tajima et al., 2016*), the mechanisms and functional impacts of the neural population-response modulation remain to be understood. To elucidate the functional impacts of neural activity modulations, the present study directly investigates the dynamical structure of the collective responses of large numbers of neurons from a decoding perspective.

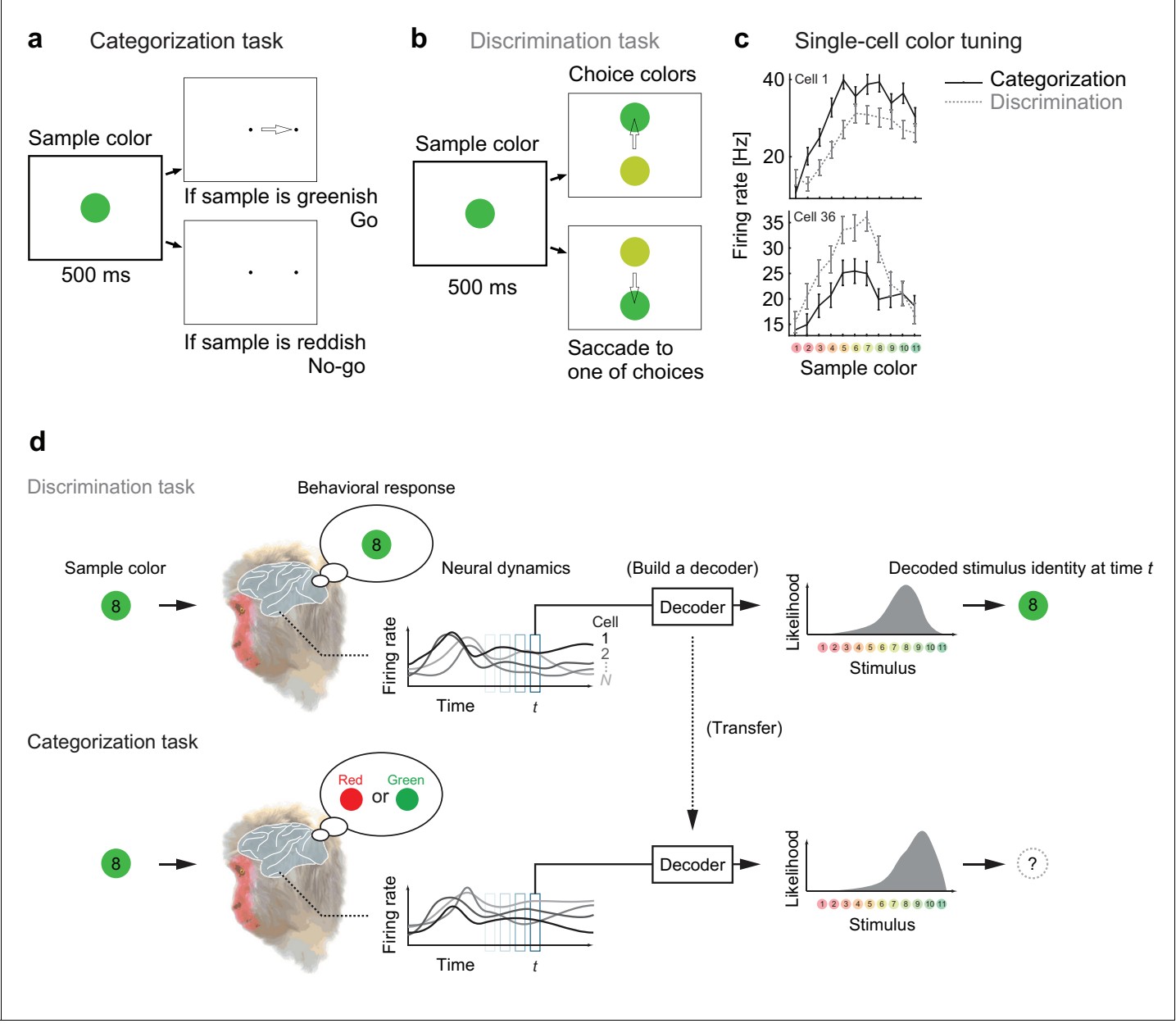

**Figure 1.** Color selectivity of IT neurons and the decoding-based stimulus reconstruction. (**a**) In the categorization task, subjects classified sample colors into either a 'reddish' or 'greenish' group. (**b**) In the discrimination task, they selected the physically identical colors. (**c**) Color tuning curves of four representative neurons in the categorization and discrimination tasks. The color selectivity and task effect varied across neurons. The average firing rates during the period spanning 100–500 ms after stimulus onset are shown. The error bars indicate the s.e.m across trials. (**d**) The likelihood-based decoding for reconstructing the stimulus representation by the neural population.

The following source data is available for figure 1:

**Source data 1.** Neural tuning data.

## Reconstructing population activity dynamics from a decoding perspective

To reconstruct the stimulus representation by the neural population, we projected the population activity to stimulus space by extending the idea of likelihood-based decoding (*Jazayeri and Movshon, 2006*; *Ma et al., 2006*; *Brouwer and Heeger, 2009*; *Graf et al., 2011*; *Fetsch et al., 2011*)

such that it captures cross-conditional differences and time-varying properties in neural population representations (Materials and methods). To obtain the joint distribution of neural activities, we generated 'pseudo-population' activities from the collection of single-neuron firing-rate distributions by randomly resampling the trials (*Fetsch et al., 2011*).We assumed no noise correlation in our main analyses although we also confirmed by additional analyses that adding noise correlation did not affect the conclusion of this study (see Discussion). The basic procedures are as follows (*Figure 1d*): to reconstruct the subjects' percepts, we first built a maximum-likelihood decoder of stimulus based on the spike-count statistics of the correct trials in the discrimination task, in which the subjects reported precise color identity during stimulus presentation; next, the same decoder was used to analyze the data from the categorization task. Note that the decoded values are matched to both the presented and the perceived stimuli in the discrimination task because we used only the correct trials from that task and the monkeys' correct rates were overall high (80–90%). Including the incorrect trials did not affect our conclusion based on the subsequent analyses. On the other hand, in the categorization task the perceived stimuli could differ from the presented stimuli. Although in the categorization task we had no access to the precise percepts of the stimulus identities but the categorical reports, we could reconstruct the putative percepts in the decoded stimulus space. The relationship between the decoder outputs and subjective percepts was also supported by follow-up analyses on the choice variability.

The decoding-based approach has two major advantages for interpreting the neural population state. First, the decoding provides a way to reduce the dimensionality of neural representation effectively by mapping the high-dimensional population state to a low-dimensional space of the perceived stimuli (which is, in the present case, one-dimensional space of color varying from red to green), which enhances visualization and analysis of the dynamical structures. Second, the decoding-based method enables clear functional interpretation of neural representation because the decoded stimuli are directly related to the subject's judgment of stimulus identity (note that it is often difficult to interpret global distance in a reduced space in nonlinear dimensionality-reduction methods; e.g., (*Roweis and Saul, 2000*; *Tenenbaum et al., 2000*; *van der Maaten and Hinton, 2008*)). In particular, the decoded stimulus identity was what the subject had to respond to in the discrimination task, and thus we can compare the decoder output and the subjects' behavior (see also Materials and methods). If the decoding is successful, it means that the neural population responses to different stimuli are effectively differentiated within the space of the decoder output. Indeed, cross-validation of the decoder performance (by dividing the data from the discrimination task into two non-overlapping sets of trials) showed a high correct rate (>75% on average across stimuli), which was comparable to the actual subject performance in the discrimination task. We will also compare the results to those of other dimensionality reduction techniques in a later subsection.

## Task context alters the attractor dynamics of the sensory neural population

To characterize the dynamical properties of the decoder output changes for the individual stimuli, we reconstructed the time evolution of the neural states within the space of decoder-output vs. mean firing rate (*Figure 2a*; the posterior given by the decoder in each stimulus and task condition is shown in *Figure 2—figure supplement 1*). The population state trajectories during the discrimination task were accurately matched the presented stimuli, confirming the successful mapping from neural representations to stimuli (*Figure 2a*, bottom). Remember that here we used the trials in which the subjects correctly identified the sample stimuli in the subsequent fine discrimination in the discrimination task (*Figure 1b*), thus the decoded stimulus identity should also correspond to the stimuli perceived by the subjects. In contrast to the discrimination task, we found that the same analysis for the Categorization task yielded strikingly different state trajectories (*Figure 2a*, top), which suggests that the neural representation was altered between the two tasks. In particular, the population state trajectories in the categorization task showed attractor-like dynamics in which the state relaxes toward either of two stable points respectively corresponding to the 'Red' and 'Green' categories along the 'line' structure (in the horizontal direction in the figure) connecting those two stable points (*Figure 2a*, top). The relationship between the mean firing rate and the decoded stimulus identity was also kept in the discrimination task and showed a similar 'line' structure with little bistability (*Figure 2a*, bottom). Interestingly, green stimuli tended to evoke larger neural responses than the red stimuli, consistently in both discrimination and categorization tasks, although the reason for

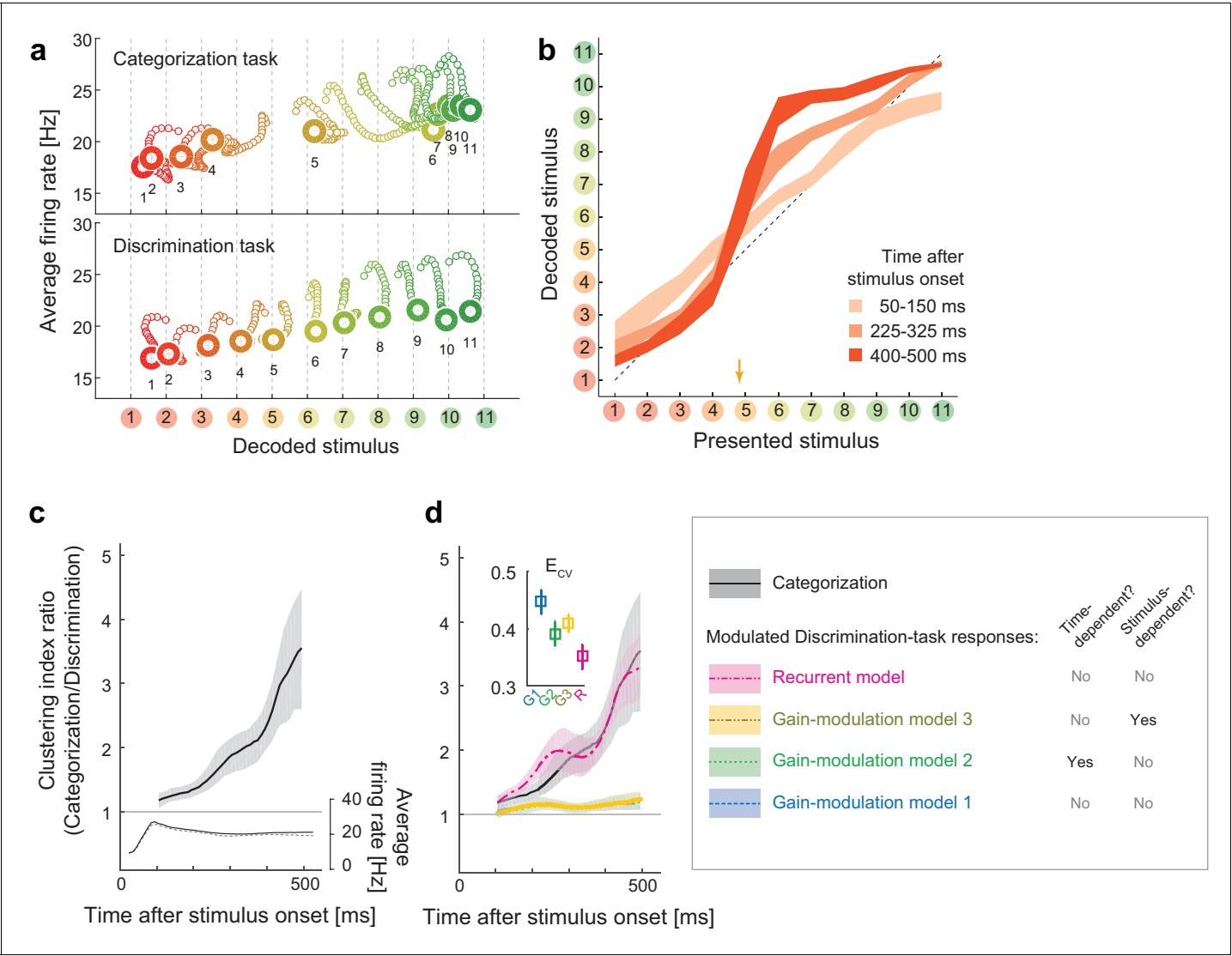

**Figure 2.** Population dynamics in the perceptual domain. (**a**) State-space trajectories during the categorization and discrimination tasks. Small markers show the population states 100–550 ms after stimulus onset in 10 ms steps. Large markers indicate the endpoint (550 ms). The colors of the trajectories and numbers around them refer to the presented stimulus. (**b**) During the categorization task, the decoded stimulus was shifted toward either the 'reddish' or 'greenish' extreme during the late responses but not during the early responses. The thickness of the curve represents the 25th–75th percentile on resampling. The yellow arrow on the horizontal axis indicates the sample color corresponding to the categorical boundary estimated from the behavior (subject's 50% response threshold) in the categorization task. (**c**) Evolution of the task-dependent clustering of the decoded stimulus (the curve with shade), as compared to the population average firing rate (the black solid and dashed curves). The magnitude of clustering was quantified with a clustering index, CI = (mean distance within categories) / (distance between category means) (Materials and methods). The figure shows the ratio of CIs in the two tasks, categorization/discrimination (smoothed with a 100 ms boxcar kernel for visualization). The horizontal line at CI ratio = 1 indicates the identity between the two tasks. The difference in CI ratio was larger in the late period (450–550 ms after the stimulus onset) than the early period (100–200 ms) (p=0.001, bootstrap test). The figure shows data averaged across all stimuli. The black curve and shaded area represent the median ±25 th percentile on the resampling. (**d**) The time-evolution of clustering indices in the gain-modulation and recurrent models applied to the discrimination-task data compared to the actual evolution in the categorization task (black curve, the same as in *Figure 3c*). The curve and shaded areas represent the median ±25 th percentile on the resampling. (Inset) The cross-validation errors ($E_{CV}$) in model fitting measured based on the individual neurons' firing rate (Materials and methods). The horizontal dashed line indicates the baseline variability in neural responses, which was quantified by comparing each neuron's firing rate in the odd and even trials. In this plot, the results of the three gain modulation models are largely overlapped with each other. Error bars: the s.e.m. across neurons. G1: gain-modulation model 1; G2: gain-modulation model 2; G3: gain-modulation model 3; R: recurrent model.

The following source data and figure supplements are available for figure 2:

**Source data 1.** Neural tuning data.

*Figure 2 continued on next page*

*Figure 2 continued*

**Figure supplement 1.** Posterior probability distributions in each stimulus and task.
**Figure supplement 2.** Trajectories and weights in trained models.

this is not clear. Finally, these properties of the dynamics were robust to various changes in the decoder construction and neural noise-correlation structures in the data, indicating that the present results do not rely on the specific designs of the decoder (see **Discussion**). We observed that the results in the eye-fixation task were similar to those of the categorization task (data not shown), replicating the previous report that the neural tunings in the eye-fixation task shared properties with the categorization task (*Koida and Komatsu, 2007*).

Remarkably, the attraction toward stable points continued throughout the stimulus presentation period, even after the population average firing rate had stabilized (as demonstrated by the horizontal shifts in *Figure 2a*, top). This also confirms that the dynamics in decoded stimuli are not merely reflecting the changes in the overall firing rate in the population (which could be potentially concerned to affect the decoding analysis through the changes in signal-to-noise ratio in the data). The polarity of the modulation depended strongly on the presented stimulus identity (*Figure 2b*). The modulation was not large at the beginning of the stimulus presentation (light plot along diagonal in *Figure 2b*) but was magnified in the late period to form the two distinct clusters (dark plots in 'S' shape, *Figure 2b*). The evolution of the task-dependent effect on clustering continued across the entire period of stimulus presentation, and was not directly associated with the dynamics of the mean firing rate, which became stable about 250 ms after the stimulus onset (*Figure 2c*).

## The recurrent model explains the stimulus-dependent dynamics

Standard models of a recurrent dynamical system, in which the system's energy function relaxes as the state evolves toward either stable point, naturally account for the dynamics converging to stable point attractors in the categorization task. In addition, the dependency on presented stimulus identity indicates that the modulation was dynamically driven by the visual input, rather than by *pre-read-out* (i.e., stimulus-invariant) modulation of neural response gains, such as conventional feature-based attention (*Treue and Martínez Trujillo, 1999*). These facts are more consistent with the recurrent model than conventional gain-modulation models as an explanation of the population dynamics reported here. To verify this, we next examined how gain-modulation and recurrent models could account for the quantitative aspects of modulation dynamics.

To analyze the dynamics of neural modulation quantitatively, we considered three gain-modulation models (in which neural response gains could depend on the task and either of time and stimulus; *Figure 2d*) and a recurrent model (response modulation via self-feedback through mutual connections to two hidden units, whose weights depended on the task but neither on time nor on the stimulus identity; *Figure 2d*). Note that we did not assume explicit stimulus-dependency of model parameters in any of the three models. We derived the model parameters based on the recorded neural responses, such that the modulated neural responses in the discrimination task fit the responses in the categorization task (full details of the modeling are provided in the Materials and methods). Using these four models, we determined to what extent the gain modulations and recurrent modulation predict the temporal evolution of decoder output changes in the categorization task. The model-fitting performances were assessed using cross-validation based on two separate sets of trials: the first set was used to train models, and the second was used to test each model's fitting performance. We computed the cross-validation errors, $E_{\mathrm{CV}}$, directly based on the difference between the predicted and actual neural population activities, thus the measure is independent of the assumptions about the decoder (Materials and methods).

Among the four models, we found that the recurrent model showed the smallest cross-validation error in terms of the individual neuron's firing rates (*Figure 2d*, inset). Indeed, neither gain-modulation model could account for the large increase in decoder output change in the late period (about >150 ms) after the stimulus onset (*Figure 2d*, the green and blue curves, the stimulus-wise

trajectories shown in *Figure 2—figure supplement 2*). The time- or stimulus-dependency of the gain parameters did not make a major difference in the prediction performance among the three gain-modulation models, suggesting that the modulation depends both on stimulus and time. On the other hand, the recurrent model explained the large continuous increase in decoder output change (*Figure 2d*, the magenta curve). It should be noted that the parameters in the recurrent model were constant across time, in contrast to the time-variant gain-modulation model. This means that the time-invariant recurrent model is superior even to the time-variant gain-modulation model at explaining the task-dependent modulation of neural population dynamics. The reason for this is that the effective modulation signals in the recurrent model could vary across different stimuli because the recurrent architecture allowed the modulation to depend on the neurons' past activities evoked by stimulus, leading to an 'implicit' dependency on stimulus and time. It is remarkable that the recurrent model is capable of describing the dynamic activity modulations without assuming any explicit parameter change across stimuli and time, even better than the time- and stimulus-dependent gain modulation, which had much more parameters than the recurrent model. The results were similar when we assumed fully-connected pairwise interactions instead of the restricted connection via the hidden units. All the results were cross-validated, making it unlikely that the difference in model performance was caused by overfitting. In addition, the superiority of the recurrent model was robustly observed with changes in the decoder construction and neural noise correlations (Discussion). These results support the idea that the task-dependency of neural dynamics originates from a recurrent mechanism, although we do not exclude the possibility of more complex gain-modulation mechanisms (that depend on both the stimulus and time) as substrates of the context-dependent dynamics observed here (see also Discussion). Note that the analysis here compares the data-fitting performance of gain-modulation and recurrent models, but does not aim at explaining how the task-dependent attractor structure can emerge. A possible mechanism underlying the task-dependent attractor dynamics is discussed later.

## Reconstructed collective dynamics explains choice variability

We also found that the neural state represented in the space of the decoded stimulus was closely related to the subjects' subsequent behavior. First, the locus of the behavioral classification boundary in the categorization task, which moderately prefers the 'Green' category, was replicated by the stimulus classification based on decoder output (*Figure 3a,b*). This suggests the decoded-stimulus space used here was closely related to the behavioral response dimension. Second, the modulation of the dynamics reflected the subjects' trial-to-trial response variability. The subject's choices between the 'Red' and 'Green' categories were variable across trials, particularly for the stimuli around the classification boundary (stimuli #4–6), even when the task condition and the presented stimulus were the same. To investigate the mechanism underlying this behavioral variability, we reanalyzed the neural responses during the categorization task using the same decoding protocol used in the previous sections, but now separated the trials into two groups according to the subsequent choice behavior. We found that the behavioral fluctuation was reflected in the preceding population dynamics in the decoded-stimulus space (*Figure 3c*). The neural state shifted toward the 'Red' extreme before the subject classified the stimulus into the 'Red' category, whereas the state shifted toward the 'Green' extreme before classifying it into the 'Green' category. The difference was small at the beginning of the response but gradually increased as time elapsed (*Figure 3d,e*). Gradual amplification of small differences in the initial state is a general property of a recurrent dynamical system having two distinct stable attractors, which further supports the recurrent model. Note that the current decoding analysis shares some concept with the conventional choice-probability analysis in single neurons (*Britten et al., 1996*), but the current decoder analysis focuses more on the collective representation by neural population. When the neuronal decoding is not linear and static, it is not necessarily straightforward to relate the dynamics of the two measures (a previous study shows modest choice probabilities at the single-neuron level in a similar color discrimination task (*Matsumora et al., 2008*)). In addition, the decoding analysis allows us to specify not only choice polarities but also the estimated perceptual contents (color identities) at each moment.

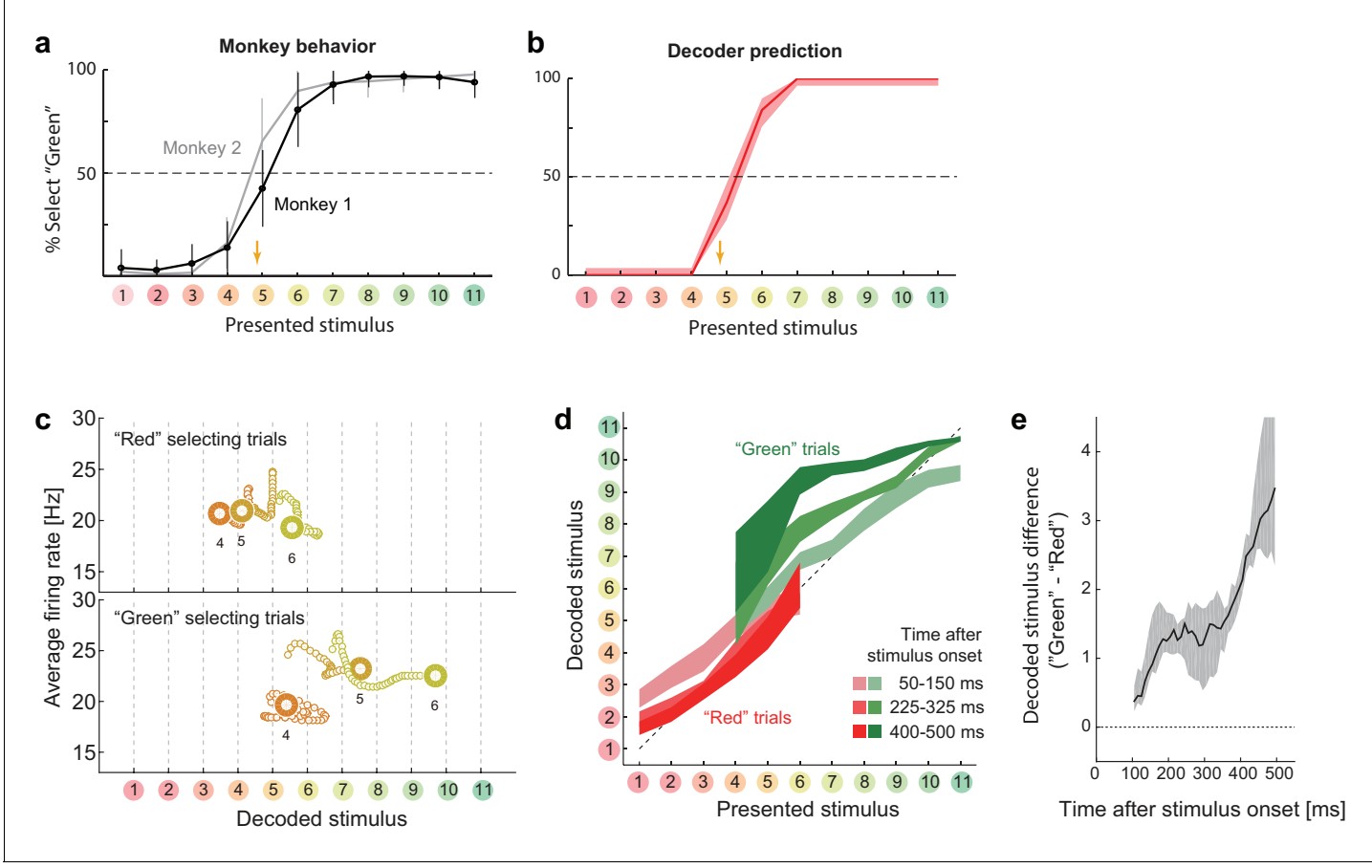

**Figure 3.** Choice-related dynamics. (a) Actual monkey behavior. Note that the monkeys' subjective category borders were consistent with the decoder output. The error bar is the standard error of mean. The yellow arrow on the horizontal axis indicates the sample color corresponding to the putative categorical boundary based on the behavior. (b) Fraction of selecting green category predicted by the likelihood-based decoding. The shaded area indicates the 25th–75th percentile on resampling. (c) The same analysis as *Figure 2a* (top) but with trial sets segregated based on whether the monkeys selected the 'red' or 'green' category. The results for stimuli #4–6 are shown. (d) The same analysis as *Figure 2b*, except that the trials were segregated based on the behavioral outcome. For stimuli #1–3 (#7–11), only the 'Red' ('Green') selecting trials were analyzed because the subjects rarely selected the other option for those stimuli. (e) Evolution of difference in the decoded color. Data were averaged across stimuli #4–6. The difference in the decoded stimulus was larger during the late period (450–550 ms) than the early period (50–150 ms) (p=0.002, permutation test).

The following source data is available for figure 3:

**Source data 1.** Neural tuning and population response data.

## Dynamical modulation enhances task-relevant information

The evidence so far indicates that the neural population in the IT cortex flexibly modulates its recurrent dynamics depending on the task context. What is this modulation for? We hypothesized that the modulation is a consequence of stimulus information processing adapted to the changing task demands. To test this possibility, we computed the mutual information between the neural population firing and the stimulus identity (hue) or stimulus category. The mutual information was estimated by assuming no noise correlations among neurons, and those estimates provide the upper limits for the information extracted from the neural state trajectories, which indicates how the dynamical modulations could contribute to the task-relevant information processing. We found that the modulatory effect was accompanied by selective increases in the task-relevant stimulus information conveyed by the neural population (*Figure 4a–c*). Namely, category information increased in the categorization task compared with the discrimination task, whereas hue information increased in the discrimination task. The difference in category information was observed in the relatively late period of the

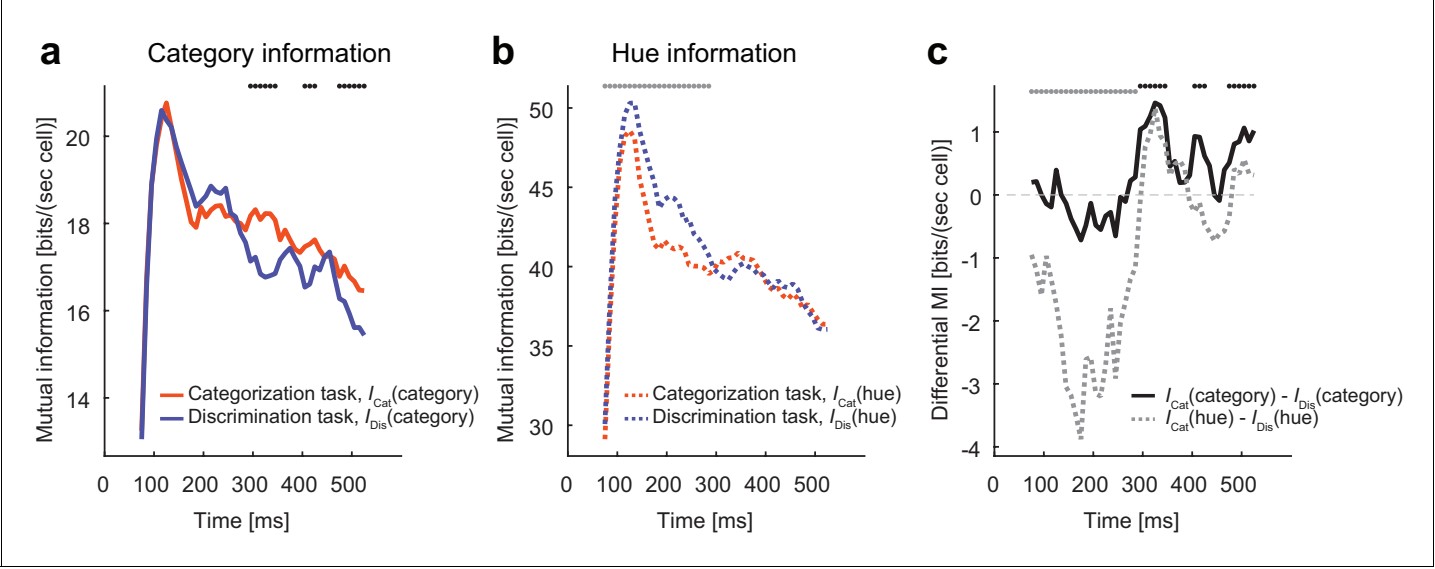

**Figure 4.** Modulation increases task-relevant information. (**a**) The evolution of mutual information about category. (**b**) The evolution of mutual information about hue. (**c**) The evolution of the mutual information difference after the stimulus onset. The dots on top of each panel indicate the statistical significance ($p<0.05$, permutation test; black dots: larger category information in the categorization task; gray dots: larger hue information in the discrimination task; the dots on top of panel c just repeat the test results shown in panels a and b).
The following source data is available for figure 4:

**Source data 1.** Neural tuning and population response data.

response (*Figure 4c*), consistent with the slow clustering dynamics described in the previous sections. On the other hand, the hue information differed only in the early period of response; this could be partially due to the stronger late responses (i.e., higher signal to noise ratio) in the categorization task. The fact that the modulation of the neural dynamics increases the task-relevant information indicates that the modulation benefits the subjects switching the tasks depending on the context.

## Comparison to other methods of dimensionality reduction

We have shown that the decoding approach captures the task-dependent attractor-like dynamics in the neural population. To examine how the other dimensionality reduction methods capture the task-dependent natures of the collective neural dynamics, we first applied the principal component analysis (PCA) to the neural responses during the stimulus presentation. *Figure 5a* shows the reconstructed trajectories and the clustering index of the neural population states in the space spanned by PCs 1–3. The trajectories for categorization and discrimination tasks largely overlapped, and the task-dependent clustering was not obvious in this space despite that these top three PCs together explained more than 60% of the total variance (*Figure 5f*). This indicates that the task-dependent components of the dynamics are hidden in the other dimensions. Similarly, it was not straightforward to demonstrate the emergence of two discrete attractors in the categorization task with other unsupervised dimensionality reduction methods such as t-stochastic neighbor embedding (tSNE) (*van der Maaten and Hinton, 2008*) (*Figure 5b*). Other semi-supervised and supervised linear dimensionality reduction/decoding methods, including the demixed PCA (*Brendel and Machens, 2011*; *Kobak et al., 2016*), a population vector (*Georgopoulos et al., 1986*) and an optimal linear decoder (*Salinas and Abbott, 1994*; *Pouget et al., 1998*), did not demonstrate the clear task-dependent clustering effect. These results implicate that the task-dependent components could be obscured when visualized naively with some of those conventional methods.

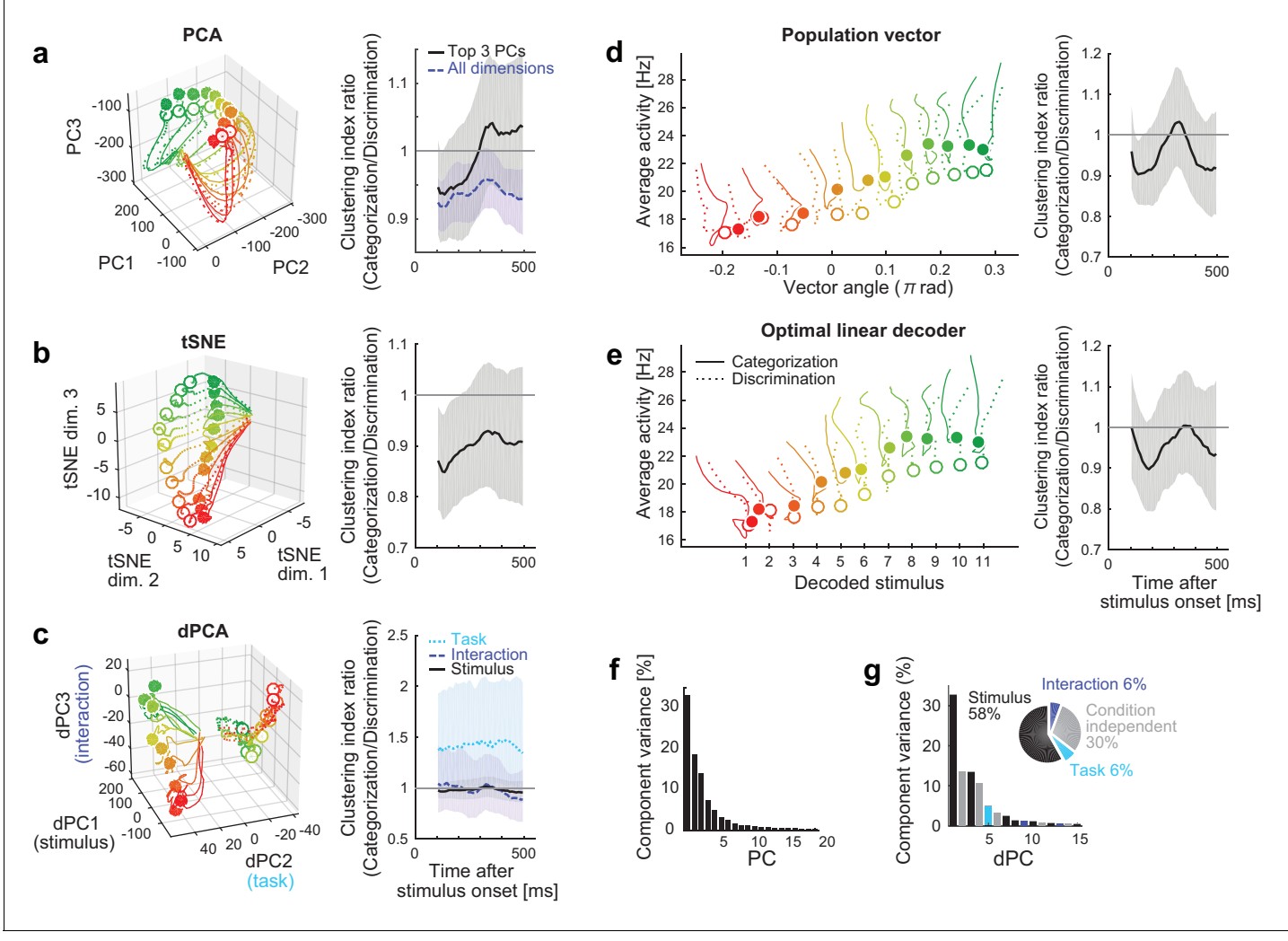

**Figure 5.** Comparison to other dimensionality reduction methods. (a–e) The results of five different dimensionality reduction methods. (Left) the trajectories visualized in the reduced state space. (Right) The time evolutions of the clustering index ratio in each reduced space. The conventions follow that of *Figure 2d*. In (a–c), the trajectories in 0–550 ms after stimulus onset are shown; in (d, e), the trajectories in 100–550 ms after stimulus onset are shown, as in the *Figure 2a*, because the decoder outputs were unreliable before 100 ms. The filled and open circles indicate the end point (at 550 ms) of the trajectories in the categorization and discrimination tasks, respectively. The colors of trajectories indicate the presented stimuli (#1-11). The parameters used in each dimensionality reduction method is provided are provided in Materials and methods. (a) PCA. The left panel shows the top 3 principal components (PCs). The left panel depicts the clustering indices based on the top 3 components (black, solid) and all the neural activities (purple, dashed). (b) Three-dimensional space obtained by t-stochastic neighbor embedding (tSNE). (c) Demixed PCA (dPCA). The left panel shows the top component from each of stimulus-dependent (black, solid), task-dependent (cyan, dotted), and stimulus-task-interaction (purple, dashed) components. (d) Population vector decoding. The horizontal and vertical axes show the decoder output and the average firing rate in population, respectively. (e) Optimal linear decoder. The horizontal and vertical axes show the decoder output and the average firing rate in population, respectively. (f) The fraction of data variance explained by each principal component (PC) in PCA. (g) The fraction of data variance explained by each demixed component (dPC) in dPCA. (Inset) The pie chart showing the relative contributions of stimulus, task, interaction and condition-independent components.

The following source data is available for figure 5:

**Source data 1.** Neural tuning and population response data.

## Bifurcation of attractor dynamics in a recurrent model

The analyses in the previous sections have indicated the flexible recurrent interactions that modulate the structures of attractors depending on the task context. What mechanism could explain such a

dynamic change in neural dynamics? Here, we show a simple potential mechanism that accounts for the flexible changes in attractor structures in the collective neural dynamics.

We extended a model of prefrontal attractor dynamics that was proposed in the context of two-interval discrimination (*Machens et al., 2005*) by introducing a recurrent interaction that involves a population of hue-selective neurons. *Figure 6a* illustrates a potential mechanism for the context-dependent change in attractor structure. We assume that the hue-selective neurons (hereafter referred to as 'hue-neurons') in the IT cortex have mutual interaction with category-selective neurons (hereafter, 'category-neurons') in the frontal or other cortical areas. The hue neurons receive sensory input from earlier visual areas. The connectivity weights between hue- and category-neurons are modeled using the functions of the preferred hues in hue-neurons such that a 'red' category-neuron exhibits excitatory interactions with hue-neurons preferring reddish hues and inhibitory interactions with neurons preferring greenish hues (similar for 'green' category-neuron). We assume that the category neurons also receive a common background input, and respond based on an activation function with response threshold and saturating nonlinearity, which characterizes the categorical response in cortical neurons (*Freedman et al., 2001*) (see Materials and methods).

This system has different numbers of stable attractors depending on the strength of common inhibitory background input (parameter $B$), with the connectivity among neurons unaffected (*Figure 6b–d*). The neural state converges to a single stable equilibrium point under a strong background inhibition (*Figure 6b*) whereas two distinct stable equilibrium points emerges under a weak or no background input, yielding bistability that depends on the initial state (*Figure 6c*). *Figure 6—figure supplement 1* shows the set of nullclines for all the color presentations, demonstrating that the bistability is observed specifically for the neutral stimuli. This agrees with the actual monkeys' behavior in which the response varied between two categories only for the neutral stimuli (#4–#6).

We confirmed that the model replicated multiple aspects of the collective neural dynamics observed in IT cortex. First, the representation of modeled hue neurons (hypothetical IT neurons) showed the gradually evolving biases toward either of two extreme stimuli ('red' or 'green'; (*Figure 6e*) as well as the moderately higher mean activity in the categorization task (*Figure 6h*). Second, the recurrent dynamics replicated the gradual development of the choice-related neural variability (*Figure 6f,g*). Third, the circuit enhanced the task-relevant information (*Figure 6i*). Finally, the task-dependent components of dynamics could be obscured when visualized with PCA (*Figure 6j*), which is also consistent with the results in IT neurons (*Figure 5a*).

## Discussion

We demonstrated that the task context modulates the structures of collective neural dynamics in the macaque IT cortex. The neural population in the IT cortex exhibited the dynamics with two discrete attractors that respectively corresponded to the two task-relevant color categories in the categorization task. The trial-to-trial variability in the dynamics confirmed that those two stable attractors co-existed under a single stimulus, thus the observed bistability reflects an inherent property of neural circuit. Remarkably, we found that the patterns of the neural state evolution were explained by a recurrent mechanism, but not fully accounted for by conventional gain-modulation models such as the ones assumed for top-down attention (*Treue and Martínez Trujillo, 1999*; *Reynolds and Heeger, 2009*).

The present hierarchical recurrent model shares some features with other recent models including the recurrent interactions between top-down and bottom-up signals (*Engel et al., 2015*; *Wimmer et al., 2015*; *Haefner et al., 2016*; *Tajima et al., 2016*). Theoretically, the hierarchical recurrent circuit can approximate a probabilistic inference on categorical stimuli, in general dynamic contexts (*Haefner et al., 2016*; *Tajima et al., 2016*). The recurrent interactions via the top-down projection are also suggested to have significant roles in learning categorical tasks (*Engel et al., 2015*) and causing the choice related fluctuations within single neuron activities (*Engel et al., 2015*; *Wimmer et al., 2015*). The present results support the model by showing the hallmarks of recurrent interactions collective neural dynamics. A unique point in the present model is that it also explains the context-dependent structure of collective neural dynamics in terms of the bifurcation of attractors caused by a simple change in the background input to the categorical neurons. Lastly, although the present results suggest a profound contribution by a recurrent mechanism to the context-dependent modulation of sensory cortex dynamics, which has not been emphasized in previous studies,

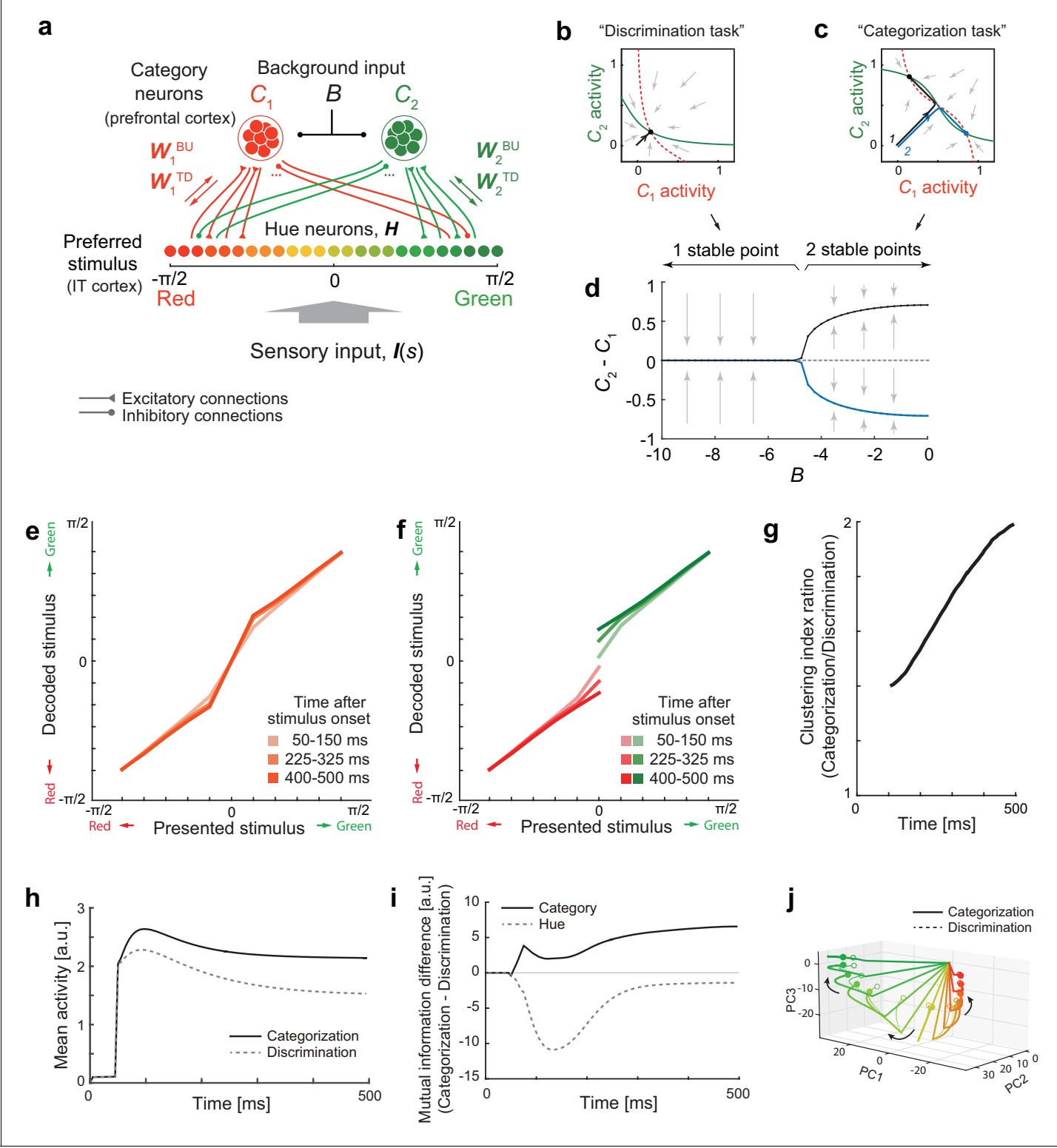

**Figure 6.** Bifurcation of attractor dynamics in a simple neural circuit model. (**a**) Schematic of the model circuit architecture. IT hue-selective neurons (hereafter, hue-neurons), $H$, with different preferred stimuli (varying from red to green) receive sensory input, $I(s)$, from the earlier visual cortex. The hue neurons interact with category neuron groups $C_1$ and $C_2$ through bottom-up and top-down connections with weights ($W_1^{BU}$, $W_2^{BU}$) and ($W_1^{TD}$, $W_2^{TD}$), respectively. The category neurons also receive a common background input, $B$, whose magnitude depends on task context. Note that the modeled hue-neurons covered entire hue circle, $[-\pi, \pi]$, although the figure shows only the half of them, corresponding to the stimulus range from red to green.

*Figure 6 continued on next page*

*Figure 6 continued*

(**b**) Activity evolution represented in the space of category-neurons in the discrimination task (where the background input $B = -8$). The red (dashed) and green (solid) curves represent nullclines for category-neurons 1 and 2, respectively. The black line shows a dynamical trajectory, starting from (0, 0) and ending at a filled circle. The gray arrows schematically illustrate the vector field. (**c**) The same analysis as in panel c but in the categorization task (where $B = -1$). The black and blue lines show two different dynamical trajectories, starting from (−0.01, 0.01) and (−0.01, 0.01), respectively (indicated as numbers '1' and '2' in the figure), and separately ending at filled circles. (**d**) The number of stable fixed points is controlled by the parameter $B$. Here, the parameter $B$ was continuously varied as the bifurcation parameter while the other parameters were kept constant. The vertical axis shows the difference of category neuron activities, $C_2 - C_1$, corresponding to the fixed points. The solid black and blue curves show the stable fixed points; the dashed line indicates the unstable fixed point. The stimulus value was $s = 0$. (**e–k**) The model replicates recorded neural population dynamics. (**e**) Presented and decoded stimuli. The same analysis as in *Figure 2b* was applied to the dynamics of the modeled hue-neurons. (**f**) The same as panel e, except that the trials were segregated based on the choices (i.e., to which fixed point the neural states were attracted). The plot corresponds to *Figure 3d*. (**g**) Evolution of difference in the decoded color, corresponding to *Figure 3e*. (**h**) Mean activity of the entire neural population, corresponding to *Figure 2c*, inset. (**i**) Differences in mutual information about category and hue between the categorization and discrimination tasks, corresponding to *Figure 4c*. (**j**) The activity trajectories of the modeled hue-neurons population in PCA space, corresponding to *Figure 5a*. Note that the scaling of the stimulus coordinate (ranging from $-\pi/2$ to $\pi/2$) used in the model is not necessarily identical to that of experimental stimuli (index by colors #1 – #11), and point of this modeling is to replicate the qualitative aspects of the data.

The following figure supplement is available for figure 6:

**Figure supplement 1.** Nullclines in different stimulus conditions and background input.

we do not exclude the potential contributions of a gain-modulation mechanism; rather, it is quite possible that the brain uses a combination of both the recurrent and feedforward mechanisms.

Recent studies emphasize a variety of stimulus-dependent contextual modulations, particularly in the early visual cortex (*Toth et al., 1996*; *Sceniak et al., 1999*, *2002*; *Sadakane et al., 2006*; *Tajima et al., 2010*; *Solomon and Kohn, 2014*; *Coen-Cagli et al., 2015*). However, it is yet to be elucidated whether the same mechanisms also apply to the context-dependent categorical processing in IT cortex as studied here, and how such a modulation could be implemented in biological systems without any recurrent mechanisms. Note that, in principle, a stimulus-dependent gain modulation requires a form of self-referencing process (which is naturally implemented by recurrent mechanisms) because it implies the stimulus encoding being modulated by the encoded stimulus itself, whether the source of modulation is the fluctuations in choice-related activity (*Nienborg and Cumming, 2009*) or attention (*Ecker et al., 2016*). Nonetheless, the mathematically equivalent effects could be achieved by a feedforward mechanism in physiological circuits that feature an information duplication (e.g., two parallel feedforward pathways converging at IT cortex, in which one has a longer latency than others). We do not exclude this possibility. This point could be tested with physiological recordings with which we can investigate the models replicating the noise correlation structures sensitive to the task contexts or estimate the causal interactions by artificially (in)activating specific areas. Our current results demonstrate that the task-dependent neural dynamics were at least not fully accounted for by conventional forms of stimulus-invariant gain modulations such as assumed in a previous study.

As a key methodology, we took a decoding approach to reconstruct the perceptual space from neural population activity. One may concern a possibility that the results rely on the selection of decoder. To examine this point, we replicated the same analyses with different decoders, and confirmed that the results reported in this paper were robust to various changes in the decoder construction, such as introducing noise correlations in neural responses, removing the half of cells to use, assuming non-Gaussian models, and ignoring the time dependence (as summarized in *Figure 7*). This suggests that the present results do not require fine tunings of the decoder constructions or assuming the independent noises across neurons. On the other hand, the task dependence of attractor structures could be unclear when visualized with-conventional unsupervised dimensionality reduction methods, despite that PCA could extract cluster structures in a previous human neuroimaging with a color naming task (*Brouwer and Heeger, 2013*). The effectiveness of the decoding approach shares some aspects with other recent labeled dimensionality-reduction approaches applied to neural population data (*Mante et al., 2013*; *Okazawa et al., 2015*). Although it is beyond the scope of the current study to compare all the possible dimensionality reduction methods, we suggest that analyzing neural-population state-space from a decoding perspective could be useful

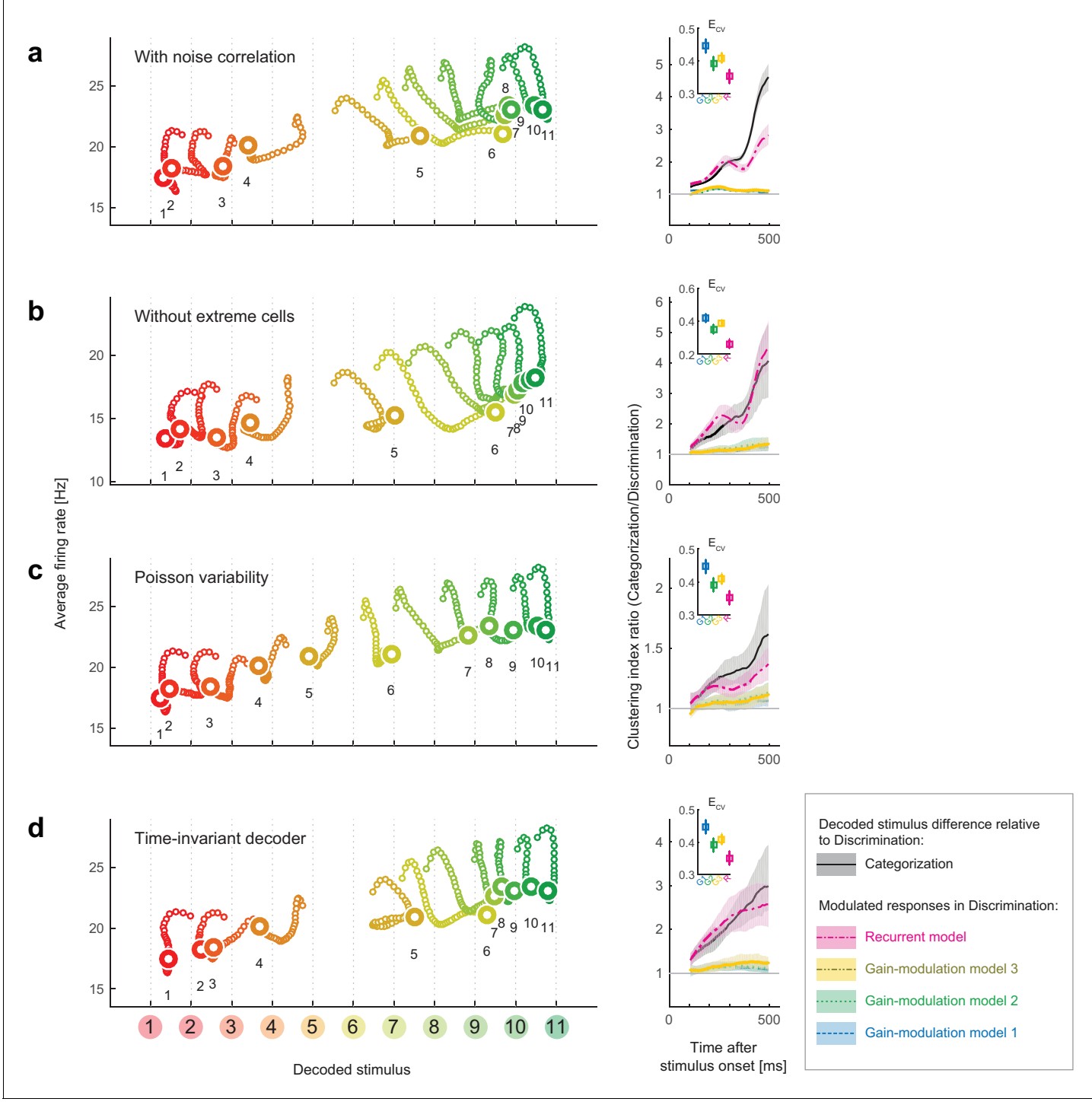

**Figure 7.** Robustness of the results to changes in the decoder. We replicated the main results of the paper using four different decoders. Both the stimulus-dependent clustering effect and the temporal evolution were replicated with those decoders. (Left) State-space trajectories during the categorization task (corresponding to *Figure 2a*, top). (Right) Time-evolution of clustering indices in the gain modulation and recurrent models compared to that in the categorization-task data (corresponding to *Figure 2d*). (a) Results obtained by simulating noise correlation among neurons. Here we assumed that the covariance $\sigma_{ij}^2$ between two different neurons, $i$ and $j$, is proportional to the correlation between their mean spike counts (*Cohen and Kohn, 2011*; *Pitkow et al., 2015*): $\sigma_{ij}^2 = k\sqrt{\alpha_i \alpha_j}\mu_i\mu_j$, where $k$ is a constant shared across all neuron pairs (here, $k = 1$), and $\alpha_i$ is the Fano factor for neuron $i$. (b) Results based on a subset of the recorded cell population; excluded are cells showing extremely high or low activity, as compared to the typical firing rate of the population. We only used cells whose average firing rates (the average across all stimuli and time bins) were within the 25th-75th percentile of the whole population. (c) Results with a decoder based on Poisson spike variability. The generative model of neuron

*Figure 7 continued on next page*

*Figure 7 continued*

*i*'s spike count in response to stimulus *s* at time *t* was given by $P(r_i(t)|s) = \mu_i(t;s)^{r_i(t)} exp(-\mu_i(t;s))/r_i(t)!$ (i.e., the log likelihood was provided by $L(s; \mathbf{r}(t)) := \log\ P(\mathbf{r}(t)|s) = \sum_{i=1}^{N} r_i(t)log\mu_i(t;s) - \sum_{i=1}^{N} \mu_i(t;s) + const.$). (**d**) Results with a time-invariant decoder. The mean and variance of each neuron's spike count were computed by pooling all the time bins during the period spanning 200–550 ms after stimulus onset.

The following source data is available for figure 7:

**Source data 1.** Neural tuning and population response data.

to extract the hidden dynamical properties that are relevant to the functions of collective neural responses.

Previous studies have proposed that context-dependent decision-making is achieved through flexible modulations of recurrent attractor dynamics within the prefrontal cortex (*Mante et al., 2013*; *Stokes et al., 2013*). The present results imply that the dynamical mechanisms of context-dependent computation can include not only the prefrontal areas but also the sensory cortex, potentially organizing the distinct representational layers such as hypothesized in the present model (*Figure 6*). Although an earlier study reported attractor-like dynamics in the IT cortex during object categorization (*Akrami et al., 2009*), the flexible modulation of a dynamical structure depending on task context has not been demonstrated. It should be noted that the present task design differs from those of many other task-switching studies: in contrast to the previous studies, in which the subjects switched behavioral rules between two different categorization tasks (e.g., categorizing motions, colors, or depths) (*Cohen and Newsome, 2008*; *Sasaki and Uka, 2009*; *Mante et al., 2013*; *Siegel et al., 2015*), the present study is based on switching between categorization and discrimination. This difference in task design may underlie the apparent discrepancy between the present and the previous studies regarding the involvement of sensory cortex in task switching. Lastly, even when no clear contextual effect is observed in single neuron properties, the correlations among neural responses can reflect the task differences (*Cohen and Newsome, 2008*), indicating that the task context affects the patterns of collective neural representations but can be missed at the single cell levels,

The way of neural modulation such that the population response becomes more sensitive to color around the categorical boundary in the categorization task is consistent with previous human psychophysics showing that the stimulus discriminability is higher around category boundaries (*Uchikawa and Sugiyama, 1993*, *1996*). Moreover, the present results add a dynamical viewpoint in neural population representations, which predicts that the perceptual illusion depends on time as well as task demands. Beyond color perception, this modulation of dynamics in sensory representation implies potential physiological substrates of task-dependent perceptual illusion. For instance, perceived motion direction is biased away from the classification boundary during a motion categorization task (*Jazayeri and Movshon, 2007*). Theoretically, this illusion could be explained both by considering direct modulation of sensory representation (*Jazayeri and Movshon, 2007*) and by assuming a readout mechanism without direct modulation of the sensory neural representation itself (*Stocker and Simoncelli, 2007*). The first model would be preferred if the motion perception is based on a population coding mechanism similar to the one demonstrated in this study which suggests the neural population representation is indeed modulated at the level of the sensory cortex.

The involvement of the sensory cortex in decision-related neural dynamics is consistent with the idea that responses within the sensory cortex are not only read out by the higher areas in a feedforward manner but also affected by decision-related signals through feedback connections from areas outside the sensory cortex (*Nienborg and Cumming, 2009*; *Siegel et al., 2015*; *Wimmer et al., 2015*). Unfortunately, we cannot fully conclude from the present data whether the observed choice-related attractor-dynamics are the *cause* or the *effect* of decision-making (*Nienborg and Cumming, 2009*). Nonetheless, the fact that modulation of the neural dynamics enhances the task-relevant information in sensory neurons may hint at the potential contribution of this modulation to the task performances. In addition, our data suggest that the choice-related difference in the dynamics had already begun during the early period (<250 ms; *Figure 2*), which is thought to affect the decision (*Nienborg and Cumming, 2009*). Therefore, it is likely that the task-dependent modulation of neural

dynamics (at least during the early period after the stimulus onset) contributed to improving the behavioral performance rather than merely reflected the decision signal. More generally, theoretical studies have proposed that a common recurrent neural circuit can serve as the basis for multiple functions, such as sensory information encoding, categorization and decision (*Wang, 2002*, *2008*; *Machens et al., 2005*; *Furman and Wang, 2008*), enabling a flexible use of the neural dynamics depending on context. The present findings suggest the involvement of sensory cortex in the context-dependent behavior, leading to a new view that the sensory neurons could contribute to context-dependent behavior by flexibly modulating their collective attractor dynamics.

## Materials and methods

### Ethics statement

This study was performed in strict accordance with the recommendations in the Guide for the Care and Use of Laboratory Animals of the National Institutes of Health. All of the animals were handled according to approved institutional animal care and use committee (IACUC) protocols of the Okazaki National Research Institutes. The protocol was approved by the Animal Experiment Committee of the Okazaki National Research Institutes (Permit Number: A16-86-29). All surgeries were performed under sodium pentobarbital anesthesia, and every effort was made to minimize suffering.

### Subjects, stimuli, and behavioral task

To study the neural basis of context-dependent behavior, we analyzed neural responses from the anterior inferior temporal (IT) cortices in two female monkeys (*Macaca fuscata*) performing visual tasks. Details of the experimental procedures have been previously published (*Koida and Komatsu, 2007*).

To study the neural basis of context-dependent behavior, we analyzed neural responses from the anterior inferior temporal (IT) cortices in two female monkeys (*Macaca fuscata*) performing visual tasks. The monkeys were trained in categorization and discrimination tasks. In both tasks, the same 11 sample colors were used as visual stimuli. The 11 sample colors ranged from red [color 1, (x = 0.631, y = 0.343 in the CIE 1931 xy chromaticity diagram)] to green [color 11, (x = 0.286, y = 0.603)] and were spaced at equal intervals on the CIE 1931 xy chromaticity diagram. The colors all had the same luminance (30 cd/m$^2$). Tasks were alternated in blocks in a fixed sequence that included categorization and discrimination tasks, as well as an eye-fixation task in which the monkey passively viewed the same color stimuli. There was no explicit cue to indicate the ongoing task. Each block consisted of 88 correct trials—eight repetitions of the 11 sample color stimuli. The 11 sample colors were presented in a pseudorandom order. If a monkey made an incorrect response to a given color, the trial using that color was repeated after some intervening trials. These repeated trials and other incomplete trials, such as those with fixation errors, were excluded from the subsequent data analyses. The stimulus was usually a disk with a diameter spanning 2.0° of visual angle, but for cells with shape selectivity, an optimal shape was chosen from among seven geometrical shapes (*Komatsu and Ideura, 1993*; *Koida and Komatsu, 2007*). The background was uniform 10 cd/m$^2$ gray (x = 0.3127, y = 0.3290). Stimuli were calibrated using a spectrophotometer (Photo Research PR-650). Personal computers controlled the task, presented the visual stimuli and recorded neural signals and eye positions. Eye movements were recorded using the scleral search coil method (*Judge et al., 1980*). The monkeys were required to maintain fixation within a 2.8° window throughout the trial, except for the saccade response. At the beginning of each trial, a small fixation spot was presented at the center of the screen. When the monkeys had gazed at the fixation spot for 500 ms, it turned off, and a sample color stimulus was presented at the center of the display for 500 ms in both categorization and discrimination tasks.

In the categorization task, after the sample stimulus was turned off, two small spots of light appeared, one at the center of the visual field, the other 5° to the right (*Figure 1a*). If the sample color belonged to the 'reddish' category (sample colors 1–4), the monkeys were rewarded for maintaining fixation on the center spot for another 700 ms ('no-go' response). If the sample color belonged to the 'greenish' category (sample colors 8–11), the monkeys were rewarded for making a saccade to the spot on the right ('go' response). For the intermediate colors (sample colors 5–7), the monkeys were rewarded randomly regardless of its behavioral response. In an early phase of the

recordings from one monkey (15 neurons), there were no intermediate colors; the 'no-go' response was assigned to colors 1–5, the 'go' response to colors 6–11.

In the discrimination task, after the sample stimulus was turned off, two choice stimuli appeared 3° above and below the fixation position (*Figure 1b*). The choice stimuli were the same shape and size as the sample stimulus; one was the same color as the sample stimulus, the other a slightly different color. The monkeys were required to make a saccade to the choice stimulus that was the same color as the sample. The two choice colors were three steps apart along the 11 sample colors – that is, the eight choice color pairs included colors #1–4, #2–5, #3–6, #4–7, #5–8, #6–9, #7–10 and #8–11. This color interval was chosen so as to yield a relatively high discriminability (about 80–90% correct). Throughout the present paper, the term 'discrimination' is used for consistency with our previous study (*Koida and Komatsu, 2007*).

## Electrophysiological recording

Neuronal activity was recorded with single unit recording from the anterior part of the IT cortex in the monkeys. We could record from 125 neurons in total. The recording region was slightly lateral to the posterior end of the anterior middle temporal sulcus (anterior 9–14 mm in the stereotaxic coordinates, area TE), which is a region where color-selective neurons are concentrated (*Komatsu et al., 1992*; *Matsumora et al., 2008*). The activities of single neurons were first isolated with online monitoring during recordings, then subject to offline spike sorting using a template matching algorithm, which confirmed that all of the data reported in this paper were single neuron activities.

All data analyses were based on neural responses to the sample colors and the fact that the monkeys saw the same visual stimuli in the categorization and discrimination tasks. For this purpose, we analyzed neural spikes recorded up to 550 ms after the sample onset, taking into account the neural response delay to the visual stimuli.

Our main results are based on a collection of single unit recordings (not a simultaneous recording of multiple neurons). In the population decoding analyses, we generated 'pseudo-population' activities from those single neuron data by randomly resampling the trials, following a procedure reported in a previous study (*Fetsch et al., 2011*). A caveat of the analysis based on 'pseudo-population' is that it omits the noise correlation (i.e., the correlation in trial-to-trial fluctuations) across neurons. As widely recognized, the noise correlation can have profound influences on the information coding by neural population, affecting particularly the resolution of sensory representation. From the decoding perspective, in many cases the noise correlation is generally considered to affect the accuracy of decoding (e.g., error bars added when plotting the decoder outputs) although how noise correlation actually limits the stimulus information is a subject of ongoing debate (*Moreno-Bote et al., 2014*). In this study, we do not primarily focus on the resolution of neural coding (reflected in the lengths of error bars) but on the 'biases' induced by the change in the mean activity in each neuron, which is captured by the present single-unit recording. In addition, a control analysis confirmed by that artificially inducing noise correlations in the studied pseudo-population did not affect the overall results (*Figure 7a*).

## Likelihood-based decoding

To visualize and characterize high-dimensional representation by neural populations, we mapped the neural population activity in the stimulus space by decoding the neural activity. From Bayes' rule, the posterior probability on stimulus $s$ under a given neural population activity $\boldsymbol{r}(t)$ is $P(s|\boldsymbol{r}(t)) \propto P(\boldsymbol{r}(t)|s)P(s)$. In a full-normative framework, the prior distribution over the stimulus could be further modeled by assuming the hierarchical model with categorical prior on stimulus, $P(s|c)$; that is, $P(s) = \int \mathrm{d}c P(s|c)P(c)$, where $c$ denotes the category information (*Tajima et al., 2016*). In the present experiments, however, the stimulus was sampled from a uniform distribution, thus the problem reduces to maximizing the likelihood $P(\boldsymbol{r}(t)|s)$. In our analysis, a maximum-likelihood decoder (*Földiák, 1991*; *Sanger, 1996*; *Jazayeri and Movshon, 2006*; *Ma et al., 2006*; *Graf et al., 2011*; *Fetsch et al., 2011*) of the stimulus was constructed based on the neural responses in the discrimination task and then applied to the data for categorization task to reconstruct the neural population states in the perceptual stimulus space (*Figure 1d*; see also the later descriptions for the rationale behind this procedure). The analyses in this and the subsequent sections are implemented using MATLAB (RRID:SCR_001622).

The decoder was constructed based on a standard likelihood-based population decoding approach as follows (*Graf et al., 2011*; *Fetsch et al., 2011*). Let $r_i(t)$ be the spike counts for the cell $i$ response at time bin $t$ in a trial. The spike count was derived from a 50 ms boxcar window whose starting point moved with 10 ms step from 0 to 500 ms after the onset of a sample-color stimulus. We first estimated a probability distribution, $P_{\mathrm{Dis}}(r_i(t)|s)$, of responses evoked by stimulus $s$ for each cell and each time bin, based on the data obtained during the discrimination task. This is approximated by a Gaussian distribution with a mean $\mu_i(t;s)$ and variance $\sigma_i(t;s)^2$, which were respectively estimated from the mean and variance in the neural spike count data. The mean responses to 11 sample stimuli ($\{\mu_i(t;1), \ldots, \mu_i(t;11)\}$) were converted to smooth functions $\mu_i(t;s)$ of the stimulus parameter $s$ (a real number varying from 1 to 11 with an interval of 0.2) through cubic interpolation (Matlab function 'interp1' with 'cubic' option) over the stimulus space, to obtain smooth likelihood functions in the later analysis. We used the smooth likelihood functions because the neural color tunings are generally considered to be continuous functions of stimulus, and the decoder outputs were more accurate (avoiding discrete jumps in trajectories) or better-interpreted by assuming the smooth tuning functions. The variance estimate was denoised by fitting a linear function, $\sigma_i(t;s)^2 = \alpha_i \times \mu_i(t;s)^2 + residual$, with a stimulus- and time-invariant scalar variable (Fano factor), $\alpha_i$, for each cell, in order to capture the potential variability in the Fano factor across neurons. To ensure that the decoder output matches the subject's perception about color identity, we used the trials in which the subjects answered correctly in the task. The Gaussian model naively implies the potential for negative neural activity, the biological meaning of which is unclear. However, this does not cause a problem in the practical data analysis because the analyzed neural responses are always positive, and we can safely equate the analysis with the one based on a rectified Gaussian model that satisfies the non-negativity of the neural responses. In addition, we also tested a Poisson distribution as a generative model of spike count, and confirmed that the results were not qualitatively affected (Results).

Combining these models of spike-count distributions derived from individual neurons and time bins yielded the log-likelihood of a population response.

$$L(s; \mathbf{r}(t)) = -(\boldsymbol{r}(t) - \mu(t;s))^\top \Sigma(t;s)^{-1} (\boldsymbol{r}(t) - \mu(t;s)) - \frac{1}{2}\log|\Sigma(t;s)| - \frac{N}{2}\log 2\pi, \tag{1}$$

where $\mu$ and $\Sigma$ are the mean and covariance of neural population response, respectively. In the main analysis, for simplicity, we assumed independent trial-to-trial variability in the neural firing (*Sanger, 1996*; *Dayan and Abbott, 2001*; *Jazayeri and Movshon, 2006*; *Ma et al., 2006*; *Brouwer and Heeger, 2009*; *Fetsch et al., 2011*)—we also observed that our main results were not affected by the decoder that takes into account the correlated variability among neurons (*Figure 7*). The joint log-likelihood $L$ of a population response of $N$ neurons, $\boldsymbol{r}(t) := (r_1(t), ..., r_N(t))^\top$, given stimulus $s$ is

$$\begin{aligned} L(s; \boldsymbol{r}(t)) :&= \log P_{\mathrm{Dis}}(\boldsymbol{r}(t)|s) \\ &= -\sum_{i=1}^{N}(r_i(t) - \mu_i(t;s))^2 / 2\sigma_i(t;s)^2 - \sum_{i=1}^{N}\log\sigma_i(t;s) - \tfrac{N}{2}\log 2\pi \end{aligned} \tag{2}$$

Here, column vector $r(t) := (r_1(t), \ldots, r_N(t))^\top$ represents the population activity of all $N$ neurons at time $t$ ($N = 125$ in the present data).. Based on this population activity, the decoder output (stimulus estimate), $s^*$, is given by maximizing the aforementioned likelihood function, $L(s; r(t))$:

$$s^*(\boldsymbol{r}(t)) := \mathrm{argmax}_s \, L(s; \boldsymbol{r}(t)). \tag{3}$$

This equation represents a mapping from the $N$-dimensional population state $r(t)$ to a one-dimensional value, $s^*$, in the stimulus space. We iterated this decoding procedure for each time $t$. We used different decoders for individual time bins, by constructing a generative model of the spike counts for each time bin. The main results were unaffected when we used a single time-invariant decoder (constructed based on the average spike count statistics across 200–550 ms after stimulus onset) for all time bins (Results).

To analyze the neural responses in the categorization task, we used the same function, $s^*(\mathbf{r}(t))$ [i. e., the same mean and variance parameters, $\left(\mu_i(t;s), \sigma_i(t;s)^2\right)$, for each neuron] as the decoder that

was constructed based on neural activity in the discrimination task. The decoder constructed based on the discrimination task data does not necessarily provide an unbiased estimate of the stimulus for the categorization task. We made use of this potential decoding bias to characterize the difference in neural population responses between the two tasks. If there were any systematic bias, it would suggest that the neural population changes the stimulus representation depending on task context. It is reasonable to construct the stimulus decoder based on neural responses in the discrimination task because the perceived stimulus identity to be decoded could be confirmed with what the subjects reported in the discrimination task. By comparing the decoder output to the subjects' behavior, we were able to map the neural population response to the subjects' perception of the stimulus identity.

The rationale behind those procedures is as follows: in the discrimination task, the subject was presented a sample color (e.g., light green), then later identified it by selecting from a pair of similar colors (e.g., the same light green vs. a slightly deeper green). When the subjects correctly identified the presented sample color, by construction, the presented color matched the chosen color, which suggests that they correctly perceived the sample color such that it could be discriminated from other similar colors in the perceptual space. Although such a correspondence between choice and perception is not always guaranteed if the subject's choice is nearly random, it was not the case in this study because the subject showed high correct rate (about 80–90%) in the discrimination task. Nonetheless, there were a few error trials in which the presented colors differed from the chosen colors. In those error trials, it is not straightforward to tell what color was perceived by the subjects; it could be the chosen color, but alternatively, they might have actually perceived the presented color but made a mistake in the response, or they might have simply unattended to the task. Thus, we excluded those error trials from the present analysis, and focused on the correct trials in which the presented and chosen colors were identical. We also confirmed that the overall results were qualitatively maintained when we replicated the same analysis using half the recorded neural population without extremely high or low activity (by eliminating the neurons showing average firing rates outside the 25th-75th percentile of the whole population; Results), which excluded the possibility that a small subset of strongly-responding neurons determined the results of decoding.

The magnitude of clustering in the decoder outputs was quantified with a clustering index, which was defined by the ratio between the mean distance within categories and the distance between category means:

$$CI(t) = \frac{\langle |s^*(t)_{|s} - s^*(t)_{|s'}| \rangle_{\, s,s' \in S_{\mathrm{Red}} \vee s,s' \in S_{\mathrm{Green}}}}{|\langle s^*(t)_{|s} \rangle_{\, s \in S_{\mathrm{Red}}} - \langle s^*(t)_{|s} \rangle_{\, s \in S_{\mathrm{Green}}}|},$$

where $S_{\mathrm{Red}} = \{\#1, \ldots, \#3\}$ and $S_{\mathrm{Green}} = \{\#6, \ldots, \#11\}$ are the 'Red' and 'Green' stimulus sets defined based on the subjects' categorization behavior.

## Fitting the task-dependent components in neural dynamics

To investigate what form of neural response modulation explains the difference between the decoded stimulus dynamics in discrimination and categorization, we fitted the population responses in the categorization task ($r(t)_{|s,\mathrm{Cat}}$) by modulating those in the discrimination task ($r(t)_{|s,\mathrm{Dis}}$), based on four different models: three feedforward gain-modulation models and a recurrent model.

### Gain modulation model 1 (time-invariant, stimulus-independent gains)

In the time-invariant gain-modulation model, the neural response data in the discrimination task were modulated so that they simulate the categorization-task responses. The simulated categorization-task response of neuron $i$, $\hat{r}_i(t)_{|s,\mathrm{Cat}}$, was provided as

$$\hat{r}_i(t)_{|s,\mathrm{Cat}} := \bar{g}_i \, r_i(t)_{|s,\mathrm{Dis}}, \tag{4}$$

where $\bar{g}_i$ denotes a constant gain-modulation for each cell $i$. The gain $\bar{g}_i$ was estimated by the linear regression (that minimizes the squared error between the predicted and actual neural responses in the categorization task). The numbers of parameters were 125 (corresponding to the number of recorded neurons $N$) in the time-invariant model. We analyzed the simulated population activity in the same procedure used for the actual response during the categorization task.

## Gain modulation model 2 (time-variant, stimulus-independent gains)

Similarly, in the time-variant gain modulation model, the predicted categorization-task response, $\hat{r}_i(t)_{|s,\text{Cat}}$, was given by

$$\hat{r}_i(t)_{|s,\text{Cat}} := g_i(t)\, r_i(t)_{|s,\text{Dis}}. \tag{5}$$

The gain term $g_i(t)$ for each neuron $i$ was estimated by the linear regression. In our main analysis, the number of neurons was $N$=125, thus the numbers of parameters were 125 $\times$51 = 6375 (corresponding to the number of recorded neurons $\times$ the number of time bins) in this model.

## Gain modulation model 3 (time-invariant, stimulus-dependent gains)

we also considered a gain modulation depending on the presented stimulus as a control (see Discussion for its biological interpretation). Note that the modulation component for each neuron can be trivially fitted by the gain modulation depending on both stimulus and time, since they are the only variables (except for the task demands) in the present experiment. Thus, here we tried to fit the data with a gain-modulation model in which the neuronal gains depend on the stimulus but not on time. In this model, the predicted categorization-task response, $\hat{r}_i(t)_{|s,\text{Cat}}$, was given by

$$\hat{r}_i(t)_{|s,\text{Cat}} := g_i(s)\, r_i(t)_{|s,\text{Dis}}. \tag{6}$$

The gain term $g_i(t)$ for each neuron $i$ was estimated by the linear regression. The numbers of model parameters were 125 $\times$11 = 1375 (corresponding to the number of recorded neurons $\times$ the number of sample colors).

## Recurrent model

Lastly, we also fitted the neural dynamics with a model that features a recurrent feedback. In the recurrent model, a self-feedback term was added to the responses in the discrimination task so that the resulting modulated activities fit those recorded in the categorization task. We assumed a restricted recurrent circuit with a single hidden layer consisting of two nonlinear hidden units. In this model, we assumed mutual connections between the recorded IT neurons and the two hidden units (which could be interpreted as the neural activity outside IT cortex, e.g., the frontal cortex, as modeled in further details later). There was no direct connection between the hidden units, resembling two-layer restricted Boltzmann machines (**Smolensky, 1986**; **Hinton, 2002**). The model is a simplified version of the circuit model (**Figure 6a**) that we used for demonstrating the task-dependent change in attractor structures (see the later section); here, we use this simplified version in the purpose of the quantitative fitting.

Based on this model, the hypothetical neural activity in the categorization task, $\hat{\boldsymbol{r}}(t)_{|s,\text{Cat}} := \left( \hat{r}_1(t)_{|s,\text{Cat}}, \ldots, \hat{r}_N(t)_{|s,\text{Cat}} \right)^{\top}$, was provided as

$$\hat{\boldsymbol{r}}(t)_{|s,\text{Cat}} := \boldsymbol{r}(t)_{|s,\text{Dis}} + \boldsymbol{Wh}(t),$$
$$\boldsymbol{h}(t) := f\left( \boldsymbol{W}^{\top} \hat{\boldsymbol{r}}(t-1)_{|s,\text{Cat}} + B \right), \tag{7}$$

where the $N \times 2$ matrix $\boldsymbol{W}$ denotes the connectivity weights between the neurons to the two hidden units. The weights are symmetric between the bottom-up and top-down connections (from the neurons to the hidden units, and from the hidden units to the neurons, respectively), as assumed commonly in the restricted Boltzmann machine (**Hinton, 2002**) and in our later simulation. The symmetric connections also avoid having the more parameters, leading to a parsimonious model. $\boldsymbol{h}(t) = (h_1(t), h_2(t))^{\top}$ is the activities of hidden units at time $t$. The function $f(\cdot) exp(x)/(1 + exp(x))$, is the activation function for the hidden units. Scalar $B$ is the common bias inputs to the hidden units. $\boldsymbol{W}$ and $B$ were learned from the data, but kept constant across time and different stimuli. To optimize those parameters, we minimized the sum of squared error between the actual and predicted neural activities in the categorization task, $\left\Vert \boldsymbol{r}(t)_{|s,\text{Cat}} - \hat{\boldsymbol{r}}(t)_{|s,\text{Cat}} \right\Vert^2$, with a standard gradient descent method on $\boldsymbol{W}$ and $B$. The number of parameters was $2N + 2 = 252$, corresponding to the total number of connections and the bias inputs. Note that it is not necessarily straightforward to relate those

two hidden units directly to the 'red' and 'green' category neurons modeled because such categorical information is represented in a mixed way in the circuit learned from the real data. Nonetheless, the goodness of fitting with this model demonstrates that the recurrent network with the restricted architecture is capable of describing the neural data quantitatively. It should be also noted that we do not consider that the task switching requires changes in all the connectivity weights among the neurons. Instead, we could assume a more parsimonious mechanism that features the attractor structure in the circuit is modulated through the change in a background input to the circuit (see the later subsection).

## Assessment of model-fitting performances

We assessed the model-fitting performances based on the cross-validation procedure as follows: we randomly divided the data into two non-overlapping sets of trials ('trial set 1' and 'trial set 2' having odd and even trial numbers, respectively), the first of which was used to train models, and the second of which was used to test each model's fitting performances. This procedure ensured that a difference in fitting performance did not reflect overfitting or a difference in the number of parameters. The model-fitting errors, $E_{\mathrm{CV}}$, were quantified by the root mean square errors between the predicted and actual neural population activities, relative to the 'baseline' variability across trials:

$$E_{\mathrm{CV}} = \frac{\left\langle \left( r_i(t)_{|s,\mathrm{Cat},\mathrm{trialset2}} - \hat{r}_i(t)_{|s,\mathrm{Cat},\mathrm{trialset2}} \right)^2 \right\rangle_{i,t,s}^{\frac{1}{2}}}{\left\langle \left( r_i(t)_{|s,\mathrm{Cat},\mathrm{trialset2}} - r_i(t)_{|s,\mathrm{Cat},\mathrm{trialset1}} \right)^2 \right\rangle_{i,t,s}^{\frac{1}{2}}} - 1, \tag{8}$$

where $\langle \cdot \rangle_{i,t,s}$ is the average over the cells, time bins, and stimuli. The numerator corresponds to the error in the model prediction, whereas the denominator represents the 'baseline' variability within the condition due to the trial-to-trial fluctuations in neural firing. Note that this measure itself is independent of the assumptions about decoders because it is computed directly from the neural population activities.

## Mutual information analysis

The amount of information about a stimulus carried by the neural population response was also evaluated using mutual information, which does not require any specific assumptions about the decoder or the models of dynamical modulations. The mutual information between the stimulus hue and the neural responses within each time bin $t$ during the categorization task was given by

$$I_{\mathrm{Cat}}(\mathrm{hue};t) = I(s,\mathbf{r}(t)) = \sum_{s,i} P_{\mathrm{Cat}}(r_i(t)|s)P(s)\{\log P_{\mathrm{Cat}}(r_i(t)|s) - \log P_{\mathrm{Cat}}(r_i(t))\}. \tag{9}$$

where $P_{\mathrm{Cat}}(r_i(t)|s)$ is the probability distribution of the $i$th neuron's response (spike counts) evoked by stimulus $s$ during the categorization task (modeled by the same normal distributions as the ones used in the likelihood-based decoding in the above). The 'hue' in the parenthesis indicates that this is the mutual information about the stimulus hue. Similarly, the mutual information between the stimulus category $c \in \{\mathrm{Red}, \mathrm{Green}\}$ and the neural responses within each time bin $t$ was given by

$$I_{\mathrm{Cat}}(\mathrm{cat};t) = I(c,\mathbf{r}(t)) = \sum_{c,i} P_{\mathrm{Cat}}(r_i(t)|c)P(c)\{\log P_{\mathrm{Cat}}(r_i(t)|c) - \log P_{\mathrm{Cat}}(r_i(t))\}, \tag{10}$$

where $P_{\mathrm{Cat}}(r_i(t)|c) = \sum_{s \in S_c} P_{\mathrm{Cat}}(r_i(t)|s)$, and $c \in \{\mathrm{Red}, \mathrm{Green}\}$ denotes the stimulus category. The 'cat' in the parenthesis indicates that this is the mutual information about the stimulus category. The mutual information values for the discrimination task, $I_{\mathrm{Dis}}(\mathrm{hue})$ and $I_{\mathrm{Dis}}(\mathrm{cat})$, were provided by substituting $P_{\mathrm{Cat}}(r_i(t)|s)$ in the above equations with the corresponding spike count distributions, $P_{\mathrm{Dis}}(r_i(t)|s)$, obtained during the discrimination task. The differential mutual information for hue and category were defined by $\Delta I(\mathrm{hue};t) = I_{\mathrm{Cat}}(\mathrm{hue};t) - I_{\mathrm{Dis}}(\mathrm{hue};t)$ and $\Delta I(\mathrm{cat};t) = I_{\mathrm{Cat}}(\mathrm{cat};t) - I_{\mathrm{Dis}}(\mathrm{cat};t)$, respectively. In the main text, we normalized the quantity by the neuron number and the time-bin length to show the information per cell and per second.

## Other dimensionality reduction analyses

We also applied five other dimensionality reduction methods to the same data and compared the results with that of the likelihood-based decoding. First, the standard principal component analysis (PCA) was applied to the set of trial-averaged data points (i.e., population response vectors $\left\{ \boldsymbol{r}(t)_{|s,\mathrm{Cat}},\ \boldsymbol{r}(t)_{|s,\mathrm{Dis}} \right\}|_{s \in \{\#1,...,\ \#11\},\ 0\ \mathrm{ms} \le t \le 550\ \mathrm{ms}}$ that varied over time $t$. Second,, we conducted t-stochastic neighbor embedding (t-SNE) (*van der Maaten and Hinton, 2008*) on the same data to examine the effects of nonlinearity in the unsupervised dimensionality reduction. Third, we performed the demixed PCA (dPCA) (*Brendel and Machens, 2011*; *Kobak et al., 2016*), which can separate the contributions of stimulus, task, and their interactions. Fourth, we applied a population vector decoding (*Georgopoulos et al., 1986*). Here, we assumed that the presented 11 stimuli span a part ($-\pi/2 \sim \pi/2$ radian angles) of a circular hue space, and decoded stimulus by $\mathrm{angle}\left( \sum_j r_j(t) e^{\mathrm{i}\phi_j} \right)$, where $\phi_j$ is neuron $j$'s preferred stimulus angle. Lastly, as a naive application of the population vector decoding is biased in non-uniform neural population, we also decoded the stimulus using a version of optimal linear decoder (*Salinas and Abbott, 1994*; *Pouget et al., 1998*), in which a readout matrix for the stimulus identity (#1–#11) was trained by the linear regression on mean responses to take into account the shapes of tuning curves and cell distribution.

## A model of context-dependent attractor dynamics

We introduced a simple recurrent model that provides a parsimonious explanation for the observed context-dependent change in attractor dynamics (see *Figure 6a*, Results). The model assumed bidirectional interactions between $n$ hue-selective neurons (hereafter, *hue neurons*) in IT cortex and two groups of category-selective neurons (*category neurons*) outside IT cortex; for example, such neurons that encode category have been found in the prefrontal cortex (*Freedman et al., 2001*; *McKee et al., 2014*). This circuit shares the basic architecture with our previous model that was proposed for general categorical inference (*Tajima et al., 2016*); here, we extend this model to explain the context dependent bifurcation of attractor dynamics. Note that the category- and hue-neurons in this model should not be confused with the terms 'categorization-' and 'discrimination-task preferred cells' used in the previous study (*Koida and Komatsu, 2007*), which were the labels on the IT neurons introduced to describe the polarity of task-dependent modulation for each cell, and not relevant to the current model.

The dynamics of category neurons were described by differential equations as follows:

$$T_C \dot{C}_1 = -C_1 + f\left( \boldsymbol{W}_1^{\mathrm{BU}} \cdot \boldsymbol{H} + B \right), \tag{11}$$

$$T_C \dot{C}_2 = -C_2 + f\left( \boldsymbol{W}_2^{\mathrm{BU}} \cdot \boldsymbol{H} + B \right), \tag{12}$$

$$\boldsymbol{H} = \boldsymbol{W}_1^{\mathrm{TD}} C_1 + \boldsymbol{W}_2^{\mathrm{TD}} C_2 + \boldsymbol{I}(s,t), \tag{13}$$

where the dots between variables denote inner products of vectors. $T_C$ is the time constant for the dynamics of category neurons, which was set as $T_C = 75$ ms in the simulation, roughly matched to the order of time constants in cortical neurons (*Murray et al., 2014*). $C_1$ and $C_2$ are scalar values representing mean activity of red- and green-preferring category neurons, respectively. The time constant for hue-neurons was neglected for the sake of the tractability in nullclines analysis. The faster dynamics in sensory neurons compared to those in higher-area is consistent with a previous report (*Murray et al., 2014*). We also confirmed that assuming non-zero time constant in hue neurons did not change the qualitative behavior of the model. The activation function in the simulation was given by a sigmoid function, $f(x) = \exp(kx)/(1 + \exp(kx))$, where $k = 0.2$, though the precise form of the activation function was not critical for the emergence of bistability as long as the neural activity was described by a monotonic saturating function. $\boldsymbol{H} := (H_1, \ldots, H_n)^\top$ is a vector representing the population activity of hue-neurons with different preferred stimuli (varying from red to green), which receive sensory input, $\boldsymbol{I}(s,t) := (I_1, \ldots, I_n)^\top$, from the earlier visual cortex. The hue neurons interact with category-neuron groups $C_1$ and $C_2$ through bottom-up and top-down connections with weights ($\boldsymbol{W}_1^{\mathrm{BU}}$, $\boldsymbol{W}_2^{\mathrm{BU}}$) and ($\boldsymbol{W}_1^{\mathrm{TD}}$, $\boldsymbol{W}_2^{\mathrm{TD}}$), respectively, where the connectivity weights were expressed as vectors (e.g.,

$\boldsymbol{W}_1^{\mathrm{BU}} := \left(W_{11}^{\mathrm{BU}}, \ldots, W_{1n}^{\mathrm{BU}}\right)^\top$). The category neurons also receive a common background input, $B$. We assume that this background input is the only component that depend on task demand in this circuit.

In the simulation, the numbers of hue-neurons were set to $n = 300$, although the size of neural population did not have major effect on the results of simulation. Sensory input to hue-neuron $i$ was modeled using a von Mises function, $I_i(s, t) = g(t) \exp\left(\kappa \, \cos\left(s - s_i^{\mathrm{pref}}\right)\right)$, where the sharpness parameter $\kappa = 2$; $s \in [-\pi/2, \pi/2]$ is the stimulus hue, which varied from red to green, and $s_i^{\mathrm{pref}}$ is the preferred hue of neuron $i$; $g(t) = 0.5e^{(t-50)/100} + 0.4$ for $t > 50$, $g(t) = 0$ for $t \leq 0$. The preferred hues were distributed uniformly across the entire hue circle, $[-\pi, \pi]$. Each category-neuron group contained 150 cells, which were uniform within each group. The connectivity weight between hue neuron $i$ and category-neuron group $j$ was modeled by $W_{ji}^{\mathrm{BU}} = W_{ji}^{\mathrm{TD}} = a\cos\left(s_j^{\mathrm{Cat}} - s_i^{\mathrm{pref}}\right)$, where $a = 10/n$, $s_j^{\mathrm{Cat}} = (-1)^j$ is the preferred hue of $C_j$. For simplicity, the bottom-up and top-down weights were assumed to be symmetric. We assumed that all the model parameters except for the background input $B$ were the same between different task conditions. The differential equations were solved with the Euler method with a unit step size of 0.25 ms.

Note that, although this model shares the basic circuit architecture with our 'first' recurrent model used for fitting the data, it includes additional model parameters to demonstrate the potential mathematical principle for the attractor bifurcation in a self-contained manner. For example, the second model includes the bottom-up inputs to the individual neurons by term $I(s, t)$, which are unknown in the actual data. (In some sense, the 'first' model used the discrimination task response as a proxy for this term). We did not fit this 'second' recurrent model directly to the data because the more model parameters can risk overfitting the data as well as make the model comparisons unstraightforward: when the model fitted the data better than the gain-modulation models, the increased model complexity can obscure whether such a model explained the data because the recurrent component was crucial or merely because it had many parameters. Nonetheless, the two models share the basic circuit architecture, and the results together demonstrate that the recurrent mechanism can describe the data as well as account for a potential mathematical principle underling the context-dependent neural dynamics.

## Acknowledgements

We thank Gouki Okazawa and Ruben Coen Cagli for helpful comments. ST was supported by JST PRESTO, Grant Number JPMJPR16E6. KA was supported by CREST, JST and JSPS KAKENHI, Grant Number 15H05707. HK was supported by the Center of Innovation Program from Japan Science and Technology Agency, JST.

## Additional information

### Funding

| Funder | Grant reference number | Author |
|---|---|---|
| Japan Science and Technology Agency | PRESTO JPMJPR16E6 | Satohiro Tajima |
| Hoso Bunka Foundation | | Satohiro Tajima |
| Japan Science and Technology Agency | CREST | Kazuyuki Aihara |
| Japan Society for the Promotion of Science | KAKENHI 15H05707 | Kazuyuki Aihara |
| Japan Science and Technology Agency | Center of Innovation Program from Japan | Hidehiko Komatsu |

The funders had no role in study design, data collection and interpretation, or the decision to submit the work for publication.

## Author contributions

ST, Conceptualization, Software, Formal analysis, Supervision, Funding acquisition, Validation, Investigation, Visualization, Methodology, Writing—original draft, Project administration, Writing—review and editing; KK, Resources, Data curation, Writing—review and editing; CIT, Writing—review and editing, Discussed the results; HS, Validation, Writing—review and editing, Discussed the results; KA, Funding acquisition, Writing—review and editing; HK, Resources, Funding acquisition, Writing—review and editing

## Author ORCIDs

Satohiro Tajima, http://orcid.org/0000-0002-9597-1381
Kowa Koida, http://orcid.org/0000-0003-0156-3406
Kazuyuki Aihara, http://orcid.org/0000-0002-4602-9816

## Ethics

Animal experimentation: This study was performed in strict accordance with the recommendations in the Guide for the Care and Use of Laboratory Animals of the National Institutes of Health. All of the animals were handled according to approved institutional animal care and use committee (IACUC) protocols of the Okazaki National Research Institutes. The protocol was approved by the Animal Experiment Committee of the Okazaki National Research Institutes (Permit Number: A16-86-29). All surgery was performed under sodium pentobarbital anesthesia, and every effort was made to minimize suffering.

# Additional files

## Supplementary files

• Source code file 1. Related to *Figure 1*: Matlab codes related to *Figure 1*. The codes run with *Figure 1—source data 1*.

• Source code file 2. Related to *Figure 2*: Matlab codes related to *Figure 2*. The codes run with *Figure 2—source data 1*.

• Source code file 3. Related to *Figure 3*: Matlab codes related to *Figure 3*. The codes run with *Figure 3—source data 1*.

• Source code file 4. Related to *Figure 4*: Matlab codes related to *Figure 4*. The codes run with *Figure 4—source data 1*.

• Source code file 5. Related to *Figure 5*: Matlab codes related to *Figure 5*. The codes run with *Figure 5—source data 1*.

• Source code file 6. Related to *Figure 6*: Matlab codes related to *Figure 6*. The codes simulate the neural population response based on the recurrent differential equation model described in the text.

• Source code file 7. Related to *Figure 7*: Matlab codes related to *Figure 7*. The codes run with *Figure 7—source data 1*.

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
