## [Decision Letter]

Thank you for submitting your article "Context-Dependent Attractor Dynamics in Visual Cortex" for consideration by *eLife*. Your article has been favorably evaluated by Timothy Behrens (Senior Editor) and three reviewers, one of whom is a member of our Board of Reviewing Editors. The reviewers have opted to remain anonymous.

The reviewers have discussed the reviews with one another and the Reviewing Editor has drafted this decision to help you prepare a revised submission.

Summary of the work:

This paper presents evidence that context-dependent dynamics occur in the sensory cortex, as opposed to the view that sensory cortex encodes the stimulus information and then passes it to the prefrontal areas for task-dependent interpretation. The authors use a very clean experimental setup, where precisely the same visual stimulus is shown regardless of task context. They then use a nonlinear decoder, which maps the neural recordings to 1D stimulus color space, to analyze the dynamics of these recordings. The decoder is trained on one task (the "Discrimination" task) and then used to interpret the results of the other task (the "Categorization" task). The decoder indicates that the dynamics in the Discrimination and Categorization task are different.

Essential revisions:

Although we have a relatively large number of comments, all should be relatively straightforward.

1) The title – Attractor Dynamics in Visual Cortex – may not be correct, and in addition may actually detract from the message of the paper. Technically, there is an attractor whenever activity goes to a fixed point. However, that's kind of trivial: given that input is fixed, we wouldn't be totally surprised if activity goes to a fixed point, so the fact that there's an attractor wouldn't be an especially big deal. It also wouldn't be an especially big deal if activity didn't go to a fixed point, as we know there are long timescales in network dynamics. And, indeed, it looks like it's the latter: while the population averaged firing rate appears to asymptote (Figure 2), the decoded stimulus is still changing at the end of the trial (Figure 2).

In any case, attractor dynamics is beside the point; what's really going on here is that there is different activity with identical input. This is a very nice illustration of task switching, but not really of attractor dynamics. That, rather than attractor dynamics, should be emphasized.

2) In the section "The recurrent model explains the stimulus-dependent dynamics," the authors look at different explanations for the differences in activity in the categorization versus discrimination tasks. The winning explanation was a recurrent model (Equation 10). All well and good, except for two things. First, the model described by Equation 10 isn't really recurrent (could have been a typo; see point 9 below). Second, later on the authors consider a different recurrent model (Equations 14-16). Why not use the second one, which seemed better to me? I suspect there's a reason, but I don't think it was well explained (either that or I missed the explanation).

3) In the section "Reconstructed collective dynamics explains choice variability," it would be a good idea to decode individual trials. If the authors really have the right model, the model should make errors on the same trial as the monkeys.

4) There are two problems with the mutual information calculation. First, it makes no sense to compute accumulated information. Even if the responses were uncorrelated across time bins, the stimulus is correlated. You can see this in the very high information differences – they're larger than the total information available. This can be partially fixed by reporting the instantaneous estimates. Here both *I*_Cat_ and *I*_Dis_ should be displayed individually, as the difference does not tell us where the information is coming from.

Second, to compute information accurately, it's necessary to have the correct noise model. And, by the author's own admission, they have the wrong noise model (they assume independent decoders). Not much they can do about this, but they should at least comment on it.

5) In the section "Comparison to other methods of dimensionality reduction," the authors claim that standard dimensionality reduction methods don't capture the difference in the discrimination versus categorization tasks. They need to do two things to back up that claim. First, they need to quantify it: differences between the trajectories in the categorization and discrimination tasks seem about as big in Figure 5 as they do in Figure 2. And in general it is more difficult to judge separation in a single 3D plot than in 1D, so a purely visual comparison isn't so useful.

Second, they should try a similarly supervised linear approach used for the nonlinear decoder. For example they could try an approach similar to the population vectors used in motor decoding, where they weight the firing rate according to a neuron's stimulus preference. An allied more modern approach is demixed PCA (Kobak et al., *eLife* 2016). Both of these differ substantially from the unsupervised PCA approach the authors use, which by comparison could be seen as a bit of a straw man for linear decoding. Although maybe not so much of a straw man; see above.

Finally, Figure 5 should also be made easier to interpret – in particular, the dots are hard to visualize.

6) In the section "Bifurcation of attractor dynamics in a recurrent model," the authors consider a second, different, recurrent model. The nullclines in Figure 6 are pretty standard. However, they are known to be somewhat fragile. Do the nullclines really keep their shape no matter what color is presented? The set of nullclines for all color presentations should be shown. It could be in Methods, and it could be all on one plot.

7) Section “Likelihood-based decoding”: Why smooth over the stimulus space? Analysis is probably easier if *s* is discrete – you can just compute the posterior for the discrete values of *s* shown to the animal. If there's a benefit to considering *s* to be a continuous variable, that should be mentioned. Otherwise, it should be discrete.

8) Something seems to be wrong with Equation 10: it's not a recurrent network; instead, r^(t) depends on r(t)andr(t−1). Is this really what the authors meant? If so, it shouldn't be called a recurrent network. If not, it needs to be corrected (there may be a hat missing somewhere).

9) The method for comparing models used in Figure 2/D, while nice for presentation as the decoder results are 1D, is difficult to interpret. Because the model prediction comes from multiple steps (first fitting the firing rates, then putting it through the decoder which could strongly transform them), it is hard to judge the meaning of the results in Figure 2. It would be good to report the mean squared error between the actual firing rates in the categorization tasks and the firing rates predicted by the model. Presumably they'll show the same trends as in panel D, but it would be nice to see that.

10) Is there a simple explanation for why the recurrent model does better? It is unclear what about the recurrent model makes it work, and some more intermediate models could be used to elucidate this. Does the recurrence matter, or is it simply fitting an additive component that works? Note that the answer may be "we can't find a simple explanation". If so, use your best judgment as to whether you want to include that in the paper.

---

## [Author Response]

*Essential revisions:*

*Although we have a relatively large number of comments, all should be relatively straightforward.*

*1) The title – Attractor Dynamics in Visual Cortex – may not be correct, and in addition may actually detract from the message of the paper. Technically, there is an attractor whenever activity goes to a fixed point. However, that's kind of trivial: given that input is fixed, we wouldn't be totally surprised if activity goes to a fixed point, so the fact that there's an attractor wouldn't be an especially big deal. It also wouldn't be an especially big deal if activity didn't go to a fixed point, as we know there are long timescales in network dynamics. And, indeed, it looks like it's the latter: while the population averaged firing rate appears to asymptote (Figure 2), the decoded stimulus is still changing at the end of the trial (Figure 2).*

*In any case, attractor dynamics is beside the point; what's really going on here is that there is different activity with identical input. This is a very nice illustration of task switching, but not really of attractor dynamics. That, rather than attractor dynamics, should be emphasized.*

We agree with the reviewers in these points, and changed the title to “Task-dependent recurrent dynamics in visual cortex.” We are open to further changes if the new title seemed not the most appropriate one.

*2) In the section "The recurrent model explains the stimulus-dependent dynamics," the authors look at different explanations for the differences in activity in the categorization versus discrimination tasks. The winning explanation was a recurrent model (Equation 10). All well and good, except for two things. First, the model described by Equation 10 isn't really recurrent (could have been a typo; see point 9 below). Second, later on the authors consider a different recurrent model (Equations 14-16). Why not use the second one, which seemed better to me? I suspect there's a reason, but I don't think it was well explained (either that or I missed the explanation).*

On the first point, we corrected the typo in old Equation 10 as the reviewer suggested (please see the response to point 8).

With regard to the second point, for a technical reason, we did not fit our ‘second’ recurrent model directly to the data. In the ‘first’ model, we needed to keep the adjustable model parameters as few as possible to avoid overfitting the data. On the other hand, the ‘second’ model includes additional parameters to demonstrate a possible mathematical principle for attractor bifurcation in a self-contained manner. For example, in the ‘second’ model, we explicitly modeled the bottom-up inputs to the individual neurons by term 𝑰(𝑠, 𝑡) (containing parameters in the number of neurons×stimuli×time bins), which were not directly accessible in the actual data. It could be theoretically possible to estimate those components from the data together with the recurrent term 𝑾 (by introducing a more advanced fitting method, such as Monte Carlo sampling). However, in practice, introducing more model parameters leads

to the overfitting/instability in fitting results, as well as make the interpretation of results unstraightforward: even when the model fitted the data well, the increased model complexity can obscure whether such a model explained the data because the recurrent component was crucial or merely because it had many parameters.

Nonetheless, the two models share the basic circuit architecture, and the results together demonstrate that the recurrent mechanism both describes the data and accounts for the potential mathematical principle underling the context-dependent neural dynamics. We explained this point in the revised manuscript (subsection “A model of context-dependent attractor dynamics”, last paragraph). In addition, in the revised manuscript, we replicated the ‘first’ model’s fitting results by making two minor modifications so that the ‘first’ model makes further matches to the ‘second’ model: using a sigmoid rather than the ‘tanh’ function for the hidden unit activation function, and assuming a single common bias input to the hidden units rather than two independent biases to them. Lastly, we also added the results showing the response trajectories generated by running the ‘first,’ trained model (Figure 2—figure supplement 2), which replicates the bistable fixed points. We expect that this would help the readers see the correspondence between the first and second models more clearly.

*3) In the section "Reconstructed collective dynamics explains choice variability," it would be a good idea to decode individual trials. If the authors really have the right model, the model should make errors on the same trial as the monkeys.*

Although we are interested in this point, unfortunately we could not do this analysis because the present dataset is not based on a simultaneous multi-unit recording but based on sequentially collected single unit recordings, from which we reconstructed the ‘virtual population activity,’ as we have described in the text (e.g., subsection “Reconstructing population activity dynamics from a decoding perspective”, first paragraph). Nonetheless, we thank the reviewers for this suggestion and would like to try this analysis when we could obtain multi-unit recording data in a future study.

*4) There are two problems with the mutual information calculation. First, it makes no sense to compute accumulated information. Even if the responses were uncorrelated across time bins, the stimulus is correlated. You can see this in the very high information differences – they're larger than the total information available. This can be partially fixed by reporting the instantaneous estimates. Here both I_Cat_ and I_Dis_ should be displayed individually, as the difference does not tell us where the information is coming from.*

*Second, to compute information accurately, it's necessary to have the correct noise model. And, by the author's own admission, they have the wrong noise model (they assume independent decoders). Not much they can do about this, but they should at least comment on it.*

We agree with both the points. We changed the plots to show the instantaneous estimates of mutual information (Figure 4), and noted that those estimates assume no noise correlations among neurons (subsection “Dynamical modulation enhances task-relevant information”). Interestingly, the increase in hue information in the Discrimination task is specific to the early period of response, which could be partially due to the relatively stronger responses (i.e., less firing variability) in the late period during the Categorization task.

*5) In the section "Comparison to other methods of dimensionality reduction," the authors claim that standard dimensionality reduction methods don't capture the difference in the discrimination versus categorization tasks. They need to do two things to back up that claim. First, they need to quantify it: differences between the trajectories in the Categorization and Discrimination tasks seem about as big in Figure 5 as they do in Figure 2. And in general it is more difficult to judge separation in a single 3D plot than in 1D, so a purely visual comparison isn't so useful.*

*Second, they should try a similarly supervised linear approach used for the nonlinear decoder. For example they could try an approach similar to the population vectors used in motor decoding, where they weight the firing rate according to a neuron's stimulus preference. An allied more modern approach is demixed PCA (Kobak et al., eLife 2016). Both of these differ substantially from the unsupervised PCA approach the authors use, which by comparison could be seen as a bit of a straw man for linear decoding. Although maybe not so much of a straw man; see above.*

*Finally, Figure 5 should also be made easier to interpret – in particular, the dots are hard to visualize.*

On the first and the second points, we quantified the task-differences in clustering effects for each dimensionality reduction method in new Figure 6. In the revised manuscript, we included the population vector decoding and demixed PCA (dPCA), as suggested. We additionally tested an optimal linear decoder, as the population vector decoding is known to be biased in non-uniform neural populations. We removed the second PCA analysis (PCA on the task-dependent component) because it is redundant when we have the result of dPCA. (For the consistency, we showed the same clustering index for the likelihood-based decoding in Figure 1/D.) These linear methods did not demonstrate clear task-dependent clustering effect (although ‘task’ component in dPCA showed some trend toward it), suggesting an importance of nonlinear methods. We also preliminary tried other discriminative models such as nonlinear support vector machines but could not obtain a stable result due to the limitation in the data size compared to the feature dimensionality.

With regard to the third point, we improved the visualization by clarifying the end point of each trajectory.

*6) In the section "Bifurcation of attractor dynamics in a recurrent model," the authors consider a second, different, recurrent model. The nullclines in Figure 6 are pretty standard. However, they are known to be somewhat fragile. Do the nullclines really keep their shape no matter what color is presented? The set of nullclines for all color presentations should be shown. It could be in Methods, and it could be all on one plot.*

As suggested, we presented the set of nullclines for a range of the stimulus colors (Figure 6—figure supplement 1). Note that the bistability is observed specifically for the neutral (‘yellowish’) stimuli, which is qualitatively consistent with the monkeys’ choice behavior (subsection “Bifurcation of attractor dynamics in a recurrent model”, third paragraph). (We presented this figure as a figure supplement because the current contents already seemed relatively dense for nontheoreticians, but we can place it to the main contents if the reviewers think it is better.)

*7) Section “Likelihood-based decoding”: Why smooth over the stimulus space? Analysis is probably easier if s is discrete – you can just compute the posterior for the discrete values of s shown to the animal. If there's a benefit to considering s to be a continuous variable, that should be mentioned. Otherwise, it should be discrete.*

The primary benefit of the smoothing is that the actual neural tuning curves are considered be continuous functions of stimulus, and we could avoid large discrete jumps in trajectories of the decoder outputs to have more robust results. We mentioned this point in the revised manuscript (subsection “Likelihood-based decoding”, second paragraph).

*8) Something seems to be wrong with Equation 10: it's not a recurrent network; instead,*
r^(t)
*depends on*
r(t)*and*r(t−1)*. Is this really what the authors meant? If so, it shouldn't be called a recurrent network. If not, it needs to be corrected (there may be a hat missing somewhere).*

This was a typo and the 𝒓 in the second equation should have been 𝒓̂ (the ‘hat’ was missing). We corrected this point in the revised manuscript.

*9) The method for comparing models used in Figure 2/D, while nice for presentation as the decoder results are 1D, is difficult to interpret. Because the model prediction comes from multiple steps (first fitting the firing rates, then putting it through the decoder which could strongly transform them), it is hard to judge the meaning of the results in Figure 2. It would be good to report the mean squared error between the actual firing rates in the categorization tasks and the firing rates predicted by the model. Presumably they'll show the same trends as in panel D, but it would be nice to see that.*

We agree that it should be better to visualize the cross-validation errors measured in the actual firing rates. We added the plot showing the cross-validation errors in the revised manuscript (Figure 2, inset).

*10) Is there a simple explanation for why the recurrent model does better? It is unclear what about the recurrent model makes it work, and some more intermediate models could be used to elucidate this. Does the recurrence matter, or is it simply fitting an additive component that works? Note that the answer may be "we can't find a simple explanation". If so, use your best judgment as to whether you want to include that in the paper.*

It might not be necessarily straightforward to provide a complete and simple/intuitive explanation for the fitting performances. Nonetheless, the better fitting performance in the recurrent model seems to come, at least partially, from the fact that it can replicate the ‘fixed point attractors’ demonstrated in Figure 2 better than the feedforward models cannot have attractors. It could be intuitively seen in the predicted neural state trajectories generated by the trained models, in which the feedforward models form less clear categorical clustering in the end points of the trajectories compared to the recurrent model (quantified in the new Figure 2; the trajectories visualized in Figure 2—figure supplement 2).

We can say at least that the better fitting performance is not merely due to the higher degree of freedom in the model (indeed, it has even less parameters than one of the feedforward models), thus we believe that the model architecture matters. We conjecture that the recurrent model works better because it describes more accurately what is happening in the actual nervous system. We also discussed possible approaches to investigating the physiological mechanisms underlying the phenomena reported in the present work (Discussion, third paragraph).